# *Colletotrichum* Species Associated with Peaches in China

**DOI:** 10.3390/jof8030313

**Published:** 2022-03-18

**Authors:** Qin Tan, Guido Schnabel, Chingchai Chaisiri, Liang-Fen Yin, Wei-Xiao Yin, Chao-Xi Luo

**Affiliations:** 1Key Lab of Horticultural Plant Biology, Ministry of Education, College of Plant Science and Technology, Huazhong Agricultural University, Wuhan 430070, China; 15207125880@163.com (Q.T.); chaisiri.ch@gmail.com (C.C.); 2Department of Plant and Environmental Sciences, Clemson University, Clemson, SC 29634, USA; schnabe@clemson.edu; 3Hubei Key Lab of Plant Pathology, College of Plant Science and Technology, Huazhong Agricultural University, Wuhan 430070, China; yh@mail.hzau.edu.cn (L.-F.Y.); wxyin@mail.hzau.edu.cn (W.-X.Y.)

**Keywords:** *Colletotrichum*, peach anthracnose, multilocus phylogeny, pathogenicity, taxonomy

## Abstract

*Colletotrichum* is regarded as one of the 10 most important genera of plant pathogens in the world. It causes diseases in a wide range of economically important plants, including peaches. China is the largest producer of peaches in the world but little is known about the *Colletotrichum* spp. affecting the crop. In 2017 and 2018, a total of 286 *Colletotrichum* isolates were isolated from symptomatic fruit and leaves in 11 peach production provinces of China. Based on multilocus phylogenetic analyses (ITS, *ACT*, *CAL*, *CHS-1*, *GAPDH*, *TUB2*, and *HIS3*) and morphological characterization, the isolates were identified to be *C. nymphaeae*, *C. fioriniae*, and *C. godetiae* of the *C. acutatum* species complex, *C. fructicola* and *C. siamense* of the *C. gloeosporioides* species complex, *C. karsti* of the *C. boninense* species complex, and one newly identified species, *C. folicola* sp. nov. This study is the first report of *C. karsti* and *C. godetiae* in peaches, and the first report of *C. nymphaeae*, *C. fioriniae*, *C. fructicola*, and *C. siamense* in peaches in China. *C. nymphaeae* is the most prevalent species of *Colletotrichum* in peaches in China, which may be the result of fungicide selection. Pathogenicity tests revealed that all species found in this study were pathogenic on both the leaves and fruit of peaches, except for *C. folicola*, which only infected the leaves. The present study substantially improves our understanding of the causal agents of anthracnose on peaches in China.

## 1. Introduction

The peach (*Prunus persica* (L.) Batsch) originated in China [1] and has been grown in many temperate climates around the world. China is the largest peach producer in the world, accounting for 55.28% of the total peach acreage in the world and 61.12% of global peach production [2]. The country produced 15,016,103 metric tons on 779,893 ha in 2020 [2].

When the temperature and humidity are favorable, *Colletotrichum* spp. can infect peaches and other fruits and cause massive economic losses [3]. *Colletotrichum* spp. pathogenic on peaches mainly infect the fruit but may also cause leaf or twig lesions. Fruit lesions appear as firm, brown, sunken (Figure 1a,c,d) areas often displaying concentric rings (Figure 1e) of small orange acervuli (Figure 1b,c,f). The acervuli produce conidia that are primarily spread by rainfall and splashing [4]. If a conidium lands on susceptible host plant tissue, it can cause secondary infection. Gumming can be observed when *Colletotrichum* spp. infect fruitlets (Figure 1a). Infected fruitlets do not reach maturity (Figure 1i), display atrophy, and eventually shrink from water loss (Figure 1i,j). Several lesions on green or mature fruit may coalesce (Figure 1a,f). *Colletotrichum* can also infect leaves with brown lesions (Figure 1g,h) and orange acervuli (Figure 1h). Severe twig infections can lead to twig dieback (Figure 1j). *Colletotrichum* species overwinter in fruit mummies and affected twigs, and form conidia in early spring [5]. In addition to asexual reproduction, they may also produce ascospores in perithecia, which were observed on apples in dead wood and on pears in leaves [6,7,8].

In the past, the taxonomy of the genus *Colletotrichum* mainly relied on host range and morphological characteristics [9]. However, these characteristics are not suitable for species-level identification since they are dependent on environmental conditions, many *Colletotrichum* species are polyphagous, and multiple species can infect the same host plant [10,11,12,13]. Molecular identification based on multilocus phylogenetic analyses or specific gene sequencing has been used for the classification and description of species concepts [3]. To date, 15 *Colletotrichum* species complexes and 22 individual species have been identified [14,15,16].

The causal agents of peach anthracnose were first reported as *Colletotrichum acutatum* and *Colletotrichum gloeosporioides* [17,18,19,20]. However, the use of molecular tools for the classification of anthracnose pathogens revealed that peach anthracnose in the USA was mostly caused by *Colletotrichum nymphaeae* and *Colletotrichum fioriniae* of the *C. acutatum* species complex [21], and *Colletotrichum siamense* and *Colletotrichum fructicola* of the *C. gloeosporioides* species complex [22]. *C. nymphaeae* was also reported in Brazil on peaches [23], and *C. fioriniae*, *C. fructicola*, and *C. siamense* were identified in South Korea on peaches [24]. Peach infections by *Colletotrichum truncatum* and *Colletotrichum acutatum* are rare [25,26].

The objective of this study was to systematically identify *Colletotrichum* spp. associated with peach fruit and leaf anthracnose in China using morphological characterization and multilocus phylogenetic analyses.

## 2. Materials and Methods

### 2.1. Isolation of Colletotrichum spp. from Peach Samples

During 2017 and 2018, the fruit and leaves of peaches with anthracnose symptoms were collected from 14 commercial peach orchards and two nurseries (Wuhan, Hubei and Fuzhou, Fujian) in 11 provinces of China, which were dry-farmed and sprayed with fungicides for anthracnose control. Conidia on diseased tissues were dipped in a cotton swab and spread on a potato dextrose agar (PDA, 20% potato infusion, 2% glucose, and 1.5% agar, and distilled water) medium and picked up with a glass needle under a professional single spore separation microscope (Wuhan Heipu Science and Technology Ltd., Wuhan, China). If no conidia were present, leaf and fruit pieces (5 × 5 mm) at the intersection of healthy and diseased tissues were surface sterilized with a sodium hypochlorite solution (1%) for 30 s and washed three times in sterilized water, followed by 75% ethanol for 30 s, then washed three times in sterilized water again. After the tissue pieces were dried, they were placed on PDA and incubated at 25 °C with a 12 h/12 h fluorescent light/dark cycle for about seven days to produce spores. Cultures were transferred to 15% diluted oatmeal agar (0.9% oatmeal, 1.5% agar, and distilled water) plates if there was no sporulation on PDA [27]. The ex-type living culture of novel species in this study was deposited in the China Center for Type Culture Collection (CCTCC), Wuhan, China.

### 2.2. Morphological Characterization

Mycelial plugs (5 mm) were transferred from the edge of actively growing cultures to fresh PDA plates and incubated at 25 °C in the dark. Colony diameters were measured after three days to calculate the mycelial growth rates (mm/d). The shape and color of colonies were investigated on the sixth day. Sexual morphs of some species were produced after four weeks. The characteristics of conidiomata were observed using fluorescence stereo microscope (Leica M205 FA, Leica Microsystem Ltd., Wetzlar, Germany). Moreover, the shape and color of conidia, conidiophores, appressoria, ascomata, asci, ascospores, and setae were recorded using a light microscope (Nikon Eclipse E400, Nikon Instruments Inc., San Francisco, CA, USA), and the length and width of 30 randomly selected conidia and 30 appressoria were measured for each representative isolate. Appressoria were induced by dropping 50 μL conidial suspension (10^5^ conidia/mL) on a microscope slide, which was placed inside a plate containing moistened filter papers with distilled water, and incubated at 25 °C in the dark for 24 to 48 h [28].

### 2.3. DNA Extraction, PCR Amplification, and Sequencing

From the 286 obtained isolates, 51 were selected for further multilocus phylogenetic analyses. They represented each geographical population, colony type, conidia morphology, and host tissue.

Fungal DNA was extracted as described previously [29]. The 5.8S nuclear ribosomal gene with the two flanking internal transcribed spacers (ITS), partial sequences of the glyceraldehyde-3-phosphate dehydrogenase gene (*GAPDH*), chitin synthase 1 gene (*CHS-1*), actin gene (*ACT*), beta-tubulin gene (*TUB2*), histone3 gene (*HIS3*), and calmodulin gene (*CAL*) were amplified and sequenced using the primer pairs described in Appendix A. The PCR conditions were 4 min at 95 °C, followed by 35 cycles of 95 °C for 30 s, annealing for 30 s at different temperatures for different genes/loci (Appendix A), and 72 °C for 45 s, with a final extension at 72 °C for 7 min. DNA sequencing was performed at Tianyi Huiyuan Biotechnology Co., Ltd. (Wuhan, China) with an ABI 3730XL sequencer from Thermo Fisher Scientific (China) Co., Ltd. (Shanghai, China). The consensus sequences were assembled from forward and reverse sequences with MEGA v. 7.0 [30]. All sequences of 51 representative *Colletotrichum* isolates in this study were submitted to GenBank and the accession numbers are listed in Appendix A.

### 2.4. Phylogenetic Analyses

Isolates were divided into four groups based on multilocus phylogenetic analyses, and type isolates of each species were selected and included in the analyses (Table 1). Multilocus phylogenetic analyses with concatenated ITS, *GAPDH*, *CHS-1*, *HIS3*, *ACT*, and *TUB2* sequences were conducted for the *C. acutatum* species complex [31]; *ACT*, *CAL*, *CHS-1*, *GAPDH*, ITS, and *TUB2* sequences were concatenated for the analysis of the *C. gloeosporioides* species complex [32]; the combined ITS, *GAPDH*, *CHS-1*, *HIS3*, *ACT*, *TUB2*, and *CAL* sequences were used to analyze the *C. boninense* species complex [33]; and the ITS, *GAPDH*, *CHS-1*, *ACT*, and *TUB2* sequences were applied for remaining species [34]. Multiple sequences were aligned and combined using MAFFT v.7 [35] and MEGA v.7.0 [30].

Bayesian inference (BI) was used to construct phylogenetic trees in MrBayes v.3.2.2 [36]. Best-fit models of nucleotide substitution were selected using MrModeltest v.2.3 [37] based on the corrected Akaike information criterion (AIC) (Table 2, Table 3, Table 4 and Table 5). BI analyses were launched with two MCMC chains that were run for 1 × 10^6^ generations (*C. acutatum* species complex and *C. boninense* species complex) [31,33], and trees sampled every 100 generations; or run 1 × 10^7^ generations (*C. gloeosporioides* species complex, and remaining species) [8,34], and trees sampled every 1000 generations. The calculation of BI analyses was stopped when the average standard deviation of split frequencies fell below 0.01. On this basis, the first 25% of generations were discarded as burn-in. Maximum parsimony (MP) analyses were implemented by using Phylogenetic Analysis Using Parsimony (PAUP*) v.4.0b10 [38]. Goodness of fit values including tree length (TL), consistency index (CI), retention index (RI), rescaled consistency index (RC), and homoplasy index (HI) were calculated for the bootstrap analyses (Table 2, Table 3, Table 4 and Table 5). Phylogenetic trees were generated using the heuristic search option with Tree Bisection Reconnection (TBR) branch swapping and 1000 random sequence additions, with all characters equally weighted and alignment gaps treated as missing data. Maximum likelihood (ML) analyses were carried out by using the CIPRES Science Gateway v.3.3 (www.phylo.org, accessed on 29 December 2021), while RAxML-HPC BlackBox was selected with default parameters. Phylogenetic trees were visualized in FigTree v.1.4.2 [39]. TreeBASE was used to store the concatenated multilocus alignments (submission number: 29227).

New species and their most closely related neighbors were analyzed using the Genealogical Concordance Phylogenetic Species Recognition (GCPSR) model by performing a pairwise homoplasy index (PHI) test [40]. The PHI test was carried out on SplitsTree v.4.14.6 [41,42] using concatenated sequences (ITS, *GAPDH*, *CHS-1*, *ACT*, and *HIS3*). The result of pairwise homoplasy index below a 0.05 threshold (Φw < 0.05) indicated the presence of significant recombination in the dataset. The relationship between closely related species was visualized by constructing a splits graph. In addition, the results of relationships between closely related species were visualized by constructing EqualAngle splits graphs, using both LogDet character transformation and split decomposition distances options.

### 2.5. Pathogenicity Test

Two to five isolates of each *Colletotrichum* sp. were used in pathogenicity tests on detached fruit and leaves. The experimental varieties for fruit and leaf inoculations were “Xiaohong” and “Xiahui No. 5”, respectively. Commercially mature fruit (still firm but with no green background color) and asymptomatic, fully developed leaves with short twigs (1–2 cm) were washed with soap and water, and surface sterilized in 1% sodium hypochlorite for 2 min and 30 s, respectively, then rinsed with sterile water and air-dried on sterile paper. Fruit was stabbed with sterilized toothpicks to produce wounds of about 5 mm deep, while leaves were punctured with sterile, medical needles. For inoculation, a 10-μL droplet of conidia suspension (1.0−2.0 × 10^5^ conidia/mL) was dropped on each wounded site, and control fruit or leaves received sterile water without conidia. Each fruit and leaf had two inoculation sites. Three fruits and three leaves were used for each isolate. Inoculated fruit and leaves were placed in a plastic tray onto 30 mm diameter plastic rings for stability. The bottom of the tray (65 cm × 40 cm × 15 cm, 24 peaches or leaves per tray) contained wet paper towels and the top was sealed with plastic film to maintain humidity. Peaches and leaves were incubated at 25 °C for six days. Pathogenicity was evaluated by the infection rates and lesion diameters. The infection rates were calculated by the formula (%) = (infected inoculation sites/all inoculation sites) × 100%. The lesion size was determined as the mean of two perpendicular diameters. The experiment was performed twice.

The fungus was re-isolated from the resulting lesions and identified as described above, thus fulfilling Koch’s postulates.

## 3. Results

From 2017 to 2018, a total of 286 *Colletotrichum* isolates were obtained from 11 provinces in China (Table 6; Figure 2a); 33 isolates were from leaves and 253 isolates were from fruit (Table 6). Although we tried to collect samples in Gansu and Shanxi provinces in northern China, no symptomatic leaves or fruit were found. *C. nymphaeae* was the most widespread and most prevalent species (Figure 2b,c), with presence in Hubei, Guizhou, Guangxi, Fujian, and Sichuan provinces. *C. fioriniae* was found in three centrally located provinces (Zhejiang, Guizhou, and Jiangxi). *C. siamense* was only found in the northernmost orchards of the collection area in Shandong and Hebei provinces, while *C. fructicola* was only found in the southernmost provinces of the collection area of Guangdong and Guizhou provinces. *C. folicola*, *C. godetiae*, and *C. karsti* were only found in Yunnan province in the westernmost border of the collection area (Table 6; Figure 2a).

### 3.1. Phylogenetic Analyses

Phylogenetic trees were constructed based on the concatenated gene/locus sequences. MP and ML trees are not shown because the topologies were similar to the displayed BI tree (Figure 3, Figure 4, Figure 5 and Figure 6). The number of taxa, aligned length (with gaps), invariable characters, uninformative variable characters, and phylogenetically informative characters of each gene/locus and combined sequences are listed in Table 2, Table 3, Table 4 and Table 5

For the *C. acutatum* species complex, in the multilocus sequence analyses (gene/locus boundaries in the alignment: ITS: 1–546, *GAPDH*: 551–815, *CHS-1*: 820–1101, *HIS3*: 1106–1492, *ACT*: 1497–1744, *TUB2*: 1749–2240) of 27 isolates from peaches in this study, 44 reference strains of *C. acutatum* species complex and one *Colletotrichum* species (*C. orchidophilum* strains CBS 632.80) as the outgroup, 2240 characters including the alignment gaps were processed. For the Bayesian analysis, a HKY + I model was selected for ITS, a HKY + G model for *GAPDH*, a K80 + I model for *CHS-1*, a GTR + I + G model for *HIS3*, and a GTR + G model for *ACT* and *TUB2*, and all were incorporated in the analysis (Table 2). As the phylogenetic tree shows in Figure 3, the 27 isolates of the *C. acutatum* species complex were clustered in three groups: 11 with *C. nymphaeae*, eight with *C. fioriniae*, and eight with *C. godetiae*. Although in the same general cluster, *C. nymphaeae* from China were genetically distinct from *C. nymphaeae* isolates from the USA and Brazil.

For the *C. gloeosporioides* species complex, DNA sequences of six genes/loci were obtained from 19 isolates from peaches in this study, with 42 reference isolates from the *C. gloeosporioides* species complex and the outgroup *C. boninense* CBS 123755. The gene/locus boundaries of the aligned 3034 characters (with gaps) were: *ACT*: 1–314, *CAL*: 319–1062, *CHS-1*: 1067–1366, *GAPDH*: 1371–1677, ITS: 1682–2295, *TUB2*: 2300–3034. For the Bayesian analysis, a HKY + G model was selected for *ACT*, a GTR + G model for *CAL*, a K80 + G model for *CHS-1*, a HKY + I model for *GAPDH* and *TUB2*, and a SYM + I + G model for ITS, and they were all incorporated in the analysis (Table 3). In the phylogenetic tree of the *C. gloeosporioides* species complex, 10 isolates clustered with *C. fructicola* and nine isolates clustered with *C. siamense* (Figure 4). They clustered together with isolates from South Korea and the USA.

Regarding the *C. boninense* species complex, in the multilocus analyses (gene/locus boundaries of ITS: 1–553, *GAPDH*: 558–843, *CHS-1*: 848–1127, *HIS3*: 1132–1524, *ACT*: 1529–1804, *TUB2*: 1809–2310, *CAL*: 2315–2763) of three isolates from peaches in this study, from 21 reference isolates of *C. boninense* species complex and one outgroup strain *C. gloeosporioides* CBS 112999, 2763 characters including the alignment gaps were processed. For the Bayesian analysis, a SYM + I + G model was selected for ITS, HKY + I for *GAPDH* and *TUB2*, K80 + G for *CHS-1*, GTR + I + G for *HIS3*, GTR + G for *ACT*, and HKY + G for *CAL*, and they were all incorporated in the analysis (Table 4). In Figure 5, three Chinese isolates clustered with *C. karsti* in the *C. boninense* species complex.

For the remaining phylogenetic analyses, the alignment of combined DNA sequences was obtained from 50 taxa, including two isolates from peaches in this study, 47 reference isolates of *Colletotrichum* species, and one outgroup strain *Monilochaetes infuscans* CBS 869.96. The gene/locus boundaries of the aligned 1981 characters (with gaps) were: ITS: 1–571, *GAPDH*: 576–896, *CHS-1*: 901–1165, *ACT*: 1170–1448, *TUB2*: 1453–1981. For the Bayesian analysis, a GTR + I + G model was selected for ITS and *CHS-1*, and HKY + I + G for *GAPDH*, *ACT*, and *TUB2*, and they were incorporated in the analysis (Table 5). In the phylogenetic tree, two isolates (YNHH2-2 and YNHH10-1 (CCTCC M 2020345)) clustered distantly from all known *Colletotrichum* species and are described herein as a new species, *C. folicola* (Figure 6). The PHI test result (Φw = 1) of *C. folicola* and its related species *C. citrus-medicae* ruled out the possibility of gene recombination interfering with the species delimitation (Figure 7). This is further evidence that *C. folicola* is a new species.

### 3.2. Taxonomy

*Colletotrichum nymphaeae* H.A. van der Aa, *Netherlands Journal of Plant Pathology*. 84: 110. (1978) (Figure 8).

Description and illustration—Damm et al. [31].

Materials examined: China, Hubei province, Yichang city, on fruit of *P. persica* cv. NJC83, April 2017, Q. Tan, living culture HBYC1; Sichuan province, Chengdu city, on fruit of *P. persica* cv. Zhongtaojinmi, June 2018, Q. Tan, living culture SCCD 1; Fujian province, Fuzhou city, on fruit of *P. persica* cv. Huangjinmi, July 2018, Q. Tan, living culture FJFZ 1; Guangxi province, Guilin city, on leaves of *P. persica* cv. Chunmei, May 2018, Q. Tan, living culture GXGL 13-1; Guizhou province, Tongren city, on fruit of *P. persica*, June 2018, Q. Tan living culture GZTR 8-1; Hubei province, Jingmen city, on fruit of *P. persica* cv. NJC83, April 2018, Q. Tan, living culture HBJM 1-1; Hubei province, Wuhan city, on fruit of *P. persica* var. *nucipersica* cv. Zhongtaojinmi, April 2017, Q. Tan, living culture HBWH 2-1; ibid, on leaves of *P. persica*, June 2017, L.F. Yin, living culture HBWH 3-2; Hubei province, Xiaogan city, on fruit of *P. persica* cv. Chunmei, May 2017, Q. Tan, living culture HBXG 1.

Notes: *Colletotrichum nymphaeae* was first described on leaves of *Nymphaea alba* in Kortenhoef by Van der Aa [43]. *C. nymphaeae* is well separated from other species with *TUB2*, but all other genes have very high intraspecific variability [31]. Consistently, *C. nymphaeae* isolates collected in this study are different from ex-type strain CBS 515.78 in ITS (2 bp), *GAPDH* (1 bp), *CHS-1* (3 bp), *ACT* (1 bp), *HIS3* (3 bp), but with 100% identity in *TUB2*.

*Colletotrichum fioriniae* (Marcelino and Gouli) R.G. Shivas and Y.P. Tan, *Fungal Diversity* 39: 117. (2009) (Figure 9).

Description and illustration—Damm et al. [31].

Materials examined: China, Jiangxi province, Jian city, on fruit of *P. persica*, August 2018, Q. Tan, living cultures JXJA 1, JXJA 6; Zhejiang province, Lishui city, on fruit of *P. persica*, September 2017, Q. Tan, living cultures ZJLS 1, ZJLS 11-1; Guizhou province, Tongren city, on fruit of *P. persica*, August 2018, Q. Tan, living culture GZTR 7-1.

Notes: *Colletotrichum acutatum* var. *fioriniae* was first isolated from *Fiorinia externa* [44] and host plants of the scale insect as an endophyte [45] in New York, USA. In 2009, Shivas and Tan identified it from *Acacia acuminate*, *Persea americana*, and *Mangifera indica* in Australia as a separate species and named it *Colletotrichum fioriniae* [46]. *C. fioriniae* was mainly isolated from wide host plants and fruits in the temperate zones [3,31]. In this study, the *C. fioriniae* isolates clustered in two subclades, which is consistent with the results of Damm’s study [31].

*Colletotrichum godetiae* P. Neergaard, *Friesia* 4: 72. (1950) (Figure 10).

Description and illustration—Damm et al. [31].

Materials examined: China, Yunnan Province, Honghe City, on leaves of *P. persica* cv. Hongxue, August 2017, Q. Tan, living cultures YNHH 1-1, YNHH 4-1, YNHH 6-1, YNHH 8-2 and YNHH 9-1.

Notes: *Colletotrichum godetiae* was first reported on the seeds of *Godetia hybrid* in Denmark by Neergaard in 1943 [47], and given detailed identification seven years later [48]. *C. godetiae* was also recovered from fruits of *Fragaria × ananassa*, *Prunus cerasus*, *Solanum betaceum*, *Citrus aurantium*, and *Olea europaea* [49]; leaves of *Laurus nobilis* and *Mahonia aquifolium*; twigs of *Ugni molinae*; and canes of *Rubus idaeus* [31]. In this study, the isolates were obtained from peach leaves and could infect both the peach fruit and leaf.

*Colletotrichum fructicola* H. Prihastuti et al., *Fungal Diversity* 39: 96. (2009) (Figure 11).

Description and illustration—Prihastuti et al. [50].

Materials examined: China, Guangdong province, Heyuan city, on fruit of *P. persica*, June 2017, Q. Tan, living culture GDHY 10-1; Guangdong province, Shaoguan city, on fruit of *P. persica* cv. Yingzuitao, August 2018, Q. Tan, living cultures GDSG 1-1, GDSG 5-1; Guizhou province, Tongren city, on fruit of *P. persica*, August 2018, Q. Tan, living culture GZTR 10-1.

Notes: *Colletotrichum fructicola* was first described from the berries of *Coffea arabica* in Chiang Mai Province, Thailand [50]. Subsequently, *C. fructicola* was reported on a wide range of hosts including *Malus domestica*, *Fragaria × ananassa*, *Limonium sinuatum*, *Pyrus pyrifolia*, *Dioscorea alata*, *Theobroma cacao Vaccinium* spp., *Vitis vinifera*, and *Prunus persica* [3,51]. In this study, the conidia and ascospores of *C. fructicola* isolates (9.3−18.9 × 3.4−8.2 µm, mean ± SD = 14.3 ± 1.7 × 5.6 ± 0.5 µm; 12.6−22.0 × 3.1–7.6 µm, mean ± SD = 17.3 ± 0.5 × 5.0 ± 0.5 µm) (Appendix A) were larger than that of ex-type (MFLU 090228, ICMP 185819: 9.7−14 × 3−4.3 µm, mean ± SD = 11.53 ± 1.03 × 3.55 ± 0.32 µm; 9−14 × 3–4 µm, mean ± SD = 11.91 ± 1.38 × 3.32 ± 0.35 µm).

*Colletotrichum siamense* H. Prihastuti et al., *Fungal Diversity* 39: 98. (2009) (Figure 12).

Description and illustration—Prihastuti et al. [50].

Materials examined: China, Shandong province, Qingdao city, on fruit of *P. persica* cv. Yangjiaomi, August 2017, Q. Tan, living cultures SDQD 1-1, SDQD 10-1; Hebei province, Shijiazhuang city, on fruit of *P. persica* cv. Dajiubao, August 2018, Q. Tan, living cultures HBSJZ 1-1, HBSJZ 3-1.

Notes: *Colletotrichum siamense* was first identified on the berries of *Coffea arabica* in Chiang Mai Province, Thailand [50] and reported to have a wide range of hosts across several tropical, subtropical, and temperate regions, including *Persea americana* and *Carica papaya* in South Africa; *Fragaria × ananassa*, *Vitis vinifera*, and *Malus domestica* in the USA; *Hymenocallis americana* and *Pyrus pyrifolia* in China; etc. [3,8,51]. In this study, we collected *C. siamense* isolates from the temperate zone in China; the conidia (13.2−18.3 × 4.6–6.3 µm, mean ± SD = 15.3 ± 0.4 × 5.4 ± 0.3 µm) (Appendix A) were larger than those of the ex-holotype (MFLU 090230, ICMP 18578: 7–18.3 × 3–4.3 µm, mean ± SD = 10.18 ± 1.74 × 3.46 ± 0.36 µm).

*Colletotrichum karsti* Y.L. Yang et al., *Cryptogamie Mycologie*. 32: 241. (2011) (Figure 13).

Description and illustration—Yang et al. [52].

Materials examined: China, Yunnan province, Honghe city, on leaves of *P. persica* cv. Hongxue, August 2017, Q. Tan, living cultures YNHH 3-1, YNHH 3-2, and YNHH 5-2.

Notes: *Colletotrichum karsti* was first described from *Vanda* sp. (*Orchidaceae*) as a pathogen on diseased leaf and endophyte of roots in Guizhou province, China [52]. *C. karsti* is the most common and geographically diverse species in the *C. boninense* species complex, and occurs on wild hosts including *Vitis vinifera*, *Capsicum* spp., *Lycopersicon esculentum*, *Coffea* sp., *Citrus* spp., *Musa banksia*, *Passiflora edulis*, *Solanum betaceum*, *Zamia obliqua*, etc. [11,33,52,53]. In this study, the conidia of *C. karsti* isolates (10.6 − 14.9 × 5.8−7.4 µm, mean ± SD = 12.9 ± 0.3 × 6.7 ± 0.2 µm) (Appendix A) were smaller than those of the ex-holotype (CGMCC3.14194: 12–19.5 × 5–7.5 µm, mean ± SD = 15.4 ± 1.3 × 6.5 ± 0.5 µm).

*Colletotrichum folicola* Q. Tan and C.X. Luo, sp. nov. (Figure 14).

MycoBank Number: MB843363.

Etymology: Referring to the host organ from which the fungus was collected.

Type: China, Yunnan Province, Honghe City, on leaves of *Prunus persica* cv. Hongxue, August 2017, Q. Tan. Holotype YNHH 10-1, Ex-type culture CCTCC M 2020345.

Sexual morphs were not observed. Asexual morphs developed on PDA. Vegetative hyphae were hyaline, smooth-walled, septate, and branched. Chlamydospores were not observed. Conidiomata acervular, conidiophores, and setae formed on hyphae or brown to black stromata. Conidiomata color ranged from yellow to grayish-yellow to light brown. Setae were medium brown to dark brown, smooth-walled, 2–6 septa, 50–140 µm long, base cylindrical, 2.5–4.5 µm in diameter at the widest part, with tip acute. Conidiophores were hyaline to pale brown, smooth-walled, septate, and up to 55 µm long. Conidiogenous cells were hyaline, cylindrical, 12.3−14.5 × 4.4–6.3 µm, with an opening of 1.8–2.5 µm. Conidia were straight, hyaline, aseptate, cylindrical, and had a round end, 12.3−15.4 × 5.6–7.8 µm, mean ± SD = 13.6 ± 0.1 × 6.5 ± 0.3 µm, L/W ratio = 2.1. Appressoria were single, dark brown, elliptical to clavate, 5.6–13.7 × 4.0−8.2 µm, mean ± SD = 8.4 ± 0.5 × 5.9 ± 0.1 µm, L/W ratio = 1.4.

Culture characteristics: Colonies on PDA attained 16–21 mm diameter in three days at 25 °C and 7–10 mm diameter in three days at 30 °C; greenish-black, white at the margin, and aerial mycelium scarce.

Additional specimens examined: China, Yunnan Province, Honghe City, on leaves of *Prunus persica* cv. Hongxue, August 2017, Q. Tan, living culture YNHH 2-2.

Notes: *Colletotrichum folicola* is phylogenetically most closely related to *C. citrus-medicae* (Figure 6). The PHI test (Φw = 1) revealed no significant recombination between *C. folicola* and *C. citrus-medicae* (Figure 7), which was described from diseased leaves of *Citrus medica* in Kunming, Yunnan Province, China [54]. *C. folicola* is different from *C. citrus-medicae* holotype isolate HGUP 1554 in ITS (with 99.04% sequence identity), *GAPDH* (99.13%), *CHS-1* (98.44%), and *HIS3* (99.72%). The sequence data of *ACT* do not separate the two species. In terms of morphology, *C. folicola* differs from *C. citrus-medicae* by having setae, smaller conidia (12.3−15.4 × 5.6−7.8 µm vs. 13.5–17 × 5.5–9 µm), longer appressoria (5.6−13.7 × 4.0−8.2 µm vs. 6–9.5 × 5.5−8.5 µm), and colonies that are greenish-black rather than white and pale brownish as in *C. citrus-medicae*.

### 3.3. Pathogenicity Tests

Pathogenicity tests were conducted to confirm Koch’s postulates on fruit and leaves for all species identified (Appendix A; Figure 15 and Figure 16). *Colletotrichum* species collected in this study showed high diversity in virulence. *C. nymphaeae*, *C. fioriniae*, *C. fructicola*, and *C. siamense*, which were already reported to be pathogens of peaches, were pathogenic on both peach leaves and fruit. *C. fructicola* and *C. siamense* from the *C. gloeosporioides* species complex were more virulent compared to species from the *C. acutatum* species complex. Interestingly, *C. folicola* and *C. karsti* showed tissue-specific pathogenicity. Isolates of these two species were all collected from leaves, and mainly infected leaves in the pathogenicity test. *C. folicola* did not infect peach fruit at all, and the size of lesions on leaves was comparably small (0.20 ± 0.06 cm). *C. karsti* did infect peach fruit, but the infection rate was only around 20% (7/36 isolates) and the size of lesions was 0.06 ± 0.01 cm. In contrast, the infection rate on leaves was 63.9% (23/36 isolates) and the lesion size was 0.35 ± 0.13 cm. Isolates of *C. godetiae* collected from peach leaves in Yunnan province were virulent on both leaves and fruit, with the leaf and fruit infection rates and lesion diameters being 88.3% (53/60 isolates) and 0.54 ± 0.05 cm and 90% (54/60 isolates) and 0.50 ± 0.17 cm, respectively (Appendix A; Figure 16).

## 4. Discussion

This study is the first large-scale investigation of *Colletotrichum* species causing anthracnose fruit and leaf diseases in peaches in China. The most common *Colletotrichum* species were *C. nymphaeae* and *C. fioriniae* of the *C. acutatum* species complex and *C. fructicola* and *C. siamense* of the *C. gloeosporioides* species complex. The same species were also identified in the southeastern USA [17,21,22], where a shift over time appeared to favor *C. gloeosporioides* species complex in South Carolina. The authors speculated that inherent resistance of *C. acutatum* to benzimidazole fungicides (MBCs) may have given this species complex a competitive advantage when MBCs were frequently used [22]. As MBCs were replaced by other fungicides (including quinone outside inhibitors and demethylation inhibitors), that competitive advantage may have disappeared and *C. gloeosporioides* species may have increased in prevalence [22,55]. In support of this hypothesis is previous research showing a higher virulence of *C. gloeosporioides* on peaches, pears, and apples compared to *C. acutatum* [8,56,57]. Also, this study and others show that the *C. gloeosporioides* species complex may be better adapted to the hot South Carolina climate compared to the *C. acutatum* species complex [3]. MBCs are still popular fungicides in Chinese peach production regions. Therefore, it is possible that the dominance of *C. acutatum* species complex, specifically *C. nymphaeae* is, at least in part, a result of fungicide selection.

The high prevalence of *C. nymphaeae* in Chinese peach orchards is consistent with other local studies reporting the same species affecting a wide variety of other fruit crops in China. For example, *C. nymphaeae* was reported in Sichuan province on blueberries and loquats [58,59], in Hubei province on strawberries and grapevines [60,61], and in Zhejiang province on pecans [62]. Internationally, it is one of the most common species affecting pome fruits, stone fruits, and small fruits [23,63,64].

*C. godetiae*, *C. karsti*, and *C. folicola* were reported on peaches for the first time. The three species were geographically isolated and only present in Yunnan province. Rare occurrences of *Colletotrichum* species have also been formerly observed on peaches, i.e., *C. truncatum* was only found in one of many orchards examined in South Carolina, USA [25]. *C. godetiae* and *C. karsti* are well-known pathogens of fruit crops. *C. godetiae* was reported to cause disease on apples, strawberries, and grapes [65,66,67,68], while *C. karsti* was reported to affect apples and blueberries [69,70]. It is, therefore, possible that these pathogens migrated from other hosts into Yunnan province peach orchards. The observed occurrence, however, does point to either a rather rare host transfer event or to environmental conditions that favor these species. Yunnan province is located in southwestern China and peach production is popular in the Yunnan–Guizhou high plateau, a region with low latitude and high altitude [71]. The complicated local topography and diverse climate lead to highly abundant biodiversity [72], which may explain the emergence of the new species *C. folicola*.

As mentioned above, regional differences in *Colletotrichum* species composition in commercial orchards may be influenced by fungicide selection pressure. For example, *C. acutatum* is less sensitive to benomyl, thiophanate-methyl, and other MBC fungicides compared with *C. gloeosporioides* [56,73,74]. Meanwhile, all *C. nymphaeae* strains in this study have been confirmed to be resistant to carbendazim (MBC) [75]. *C. nymphaeae* was reported to be less sensitive to demethylation inhibitor (DMIs) fungicides (flutriafol and fenbuconazole) compared with *C. fioriniae*, *C. fructicola*, and *C. siamense* [21] and *C. gloeosporioides* was reported to be inherently tolerant to fludioxonil [76,77]. Most of the peach farms in China are small and there is vast diversity in the approaches to managing diseases. However, MBC (i.e., carbendazim and thiophanate-methyl) fungicides are commonly used to control peach diseases, followed by DMIs (i.e., difenoconazole). Whether fungicide selection had an impact on the *Colletotrichum* species distribution is unknown, but the high prevalence of *C. acutatum* species complex and their resilience to MBCs (and, in the case of *C. nymphaeae*, to DMIs) would allow for such a hypothesis.

In conclusion, this study provides the morphological, molecular, and pathological characterization of seven *Colletotrichum* spp. occurring on peaches in China. This is of great significance for the prevention and control of anthracnose disease in different areas in China.

## Figures and Tables

**Figure 1 jof-08-00313-f001:**
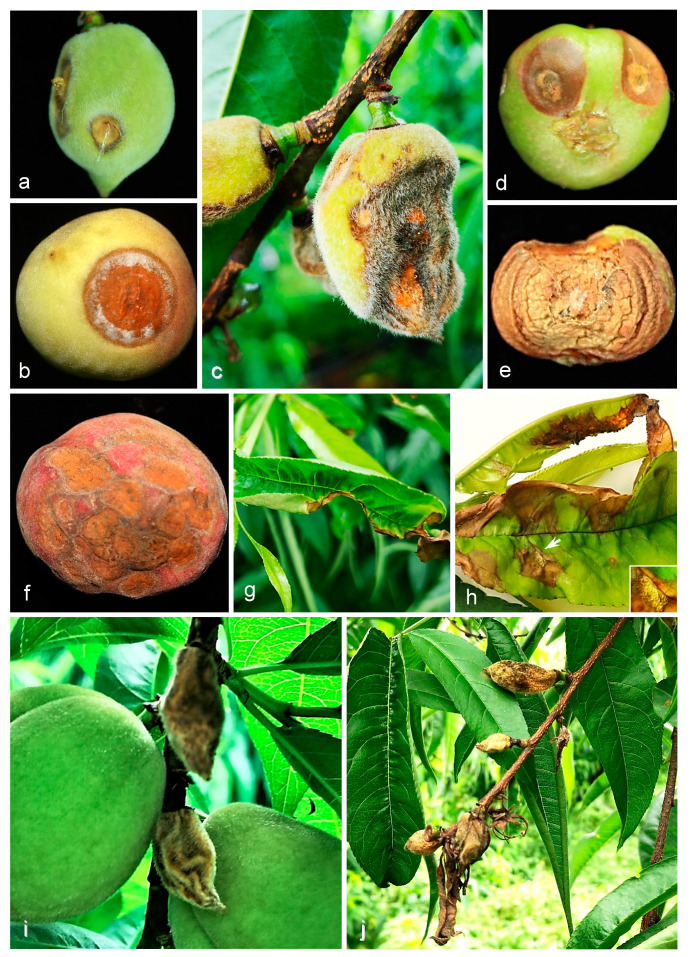
Symptoms of peach anthracnose on fruit and leaves. (**a**–**f**) Various symptoms on fruit of *Prunus persica* (**a**–**c**,**f**) and *P. persica* var. *nucipersica* (**d**,**e**): (**a**,**c**–**e**) lesions on fruitlets and (**b**,**f**) lesions on mature peach fruit; (**g**,**h**) anthracnose symptoms on leaves; (**i**) mumified young fruit; (**j**) infected twig.

**Figure 2 jof-08-00313-f002:**
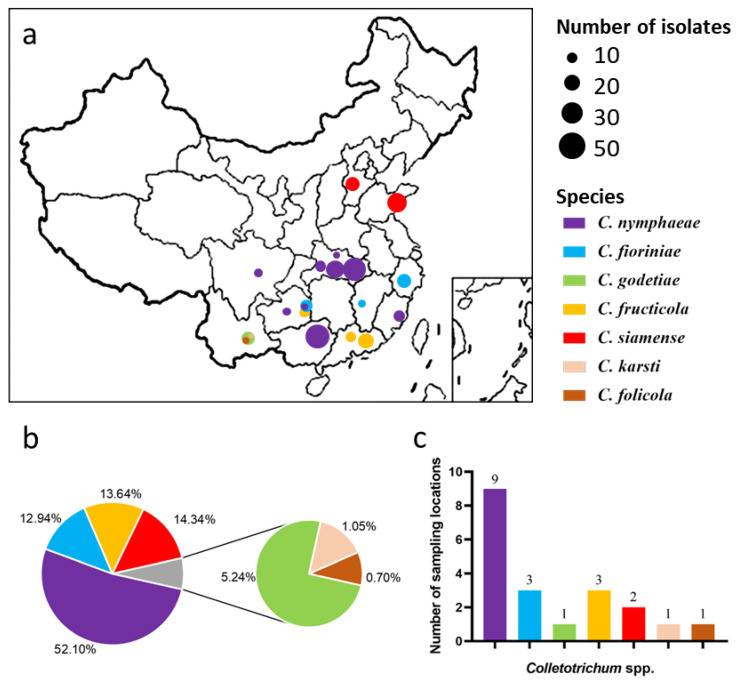
Prevalence of *Colletotrichum* spp. associated with peaches in China. (**a**) Map of the distribution of *Colletotrichum* spp. on peaches in China. Each color represents one *Colletotrichum* species, and the size of the circle indicates the number of isolates collected from that location. (**b**) Overall isolation rate (%) of *Colletotrichum* species; (**c**) number of sampling locations for each *Colletotrichum* species.

**Figure 3 jof-08-00313-f003:**
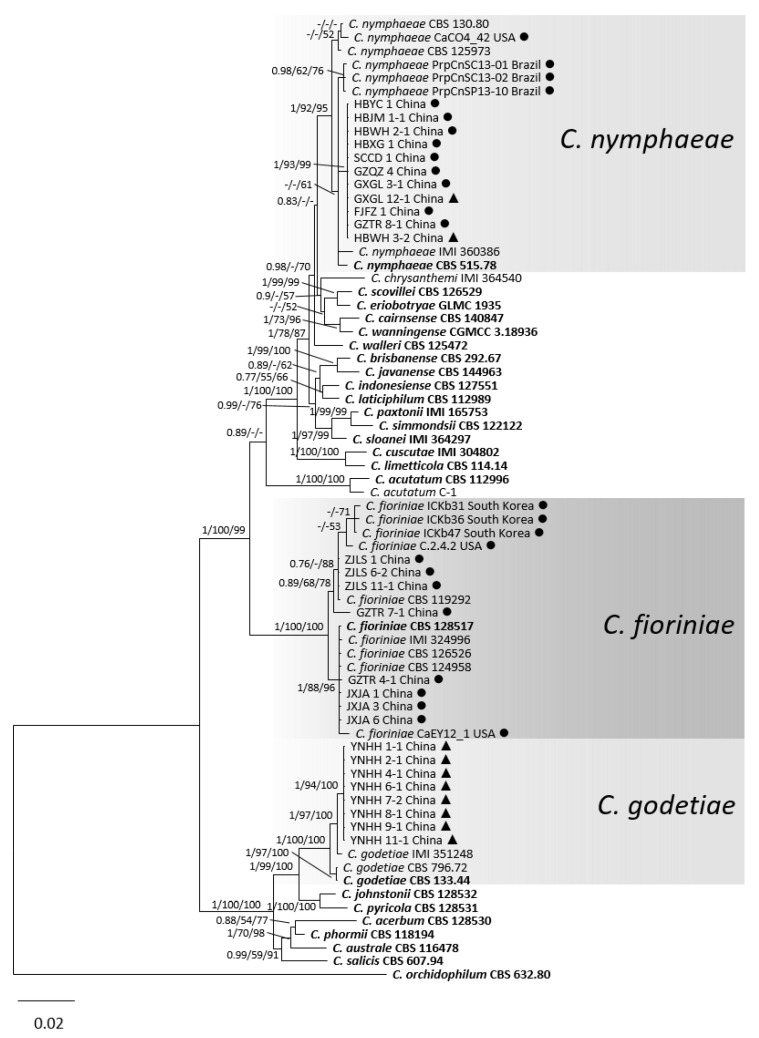
A Bayesian inference phylogenetic tree of 71 isolates in the *C. acutatum* species complex. *C. orchidophilum* (CBS 632.80) was used as the outgroup. The tree was built using combined sequences of the ITS, *GAPDH*, *CHS-1*, *HIS3*, *ACT*, and *TUB2*. BI posterior probability values (BI ≥ 0.70), MP bootstrap support values (MP ≥ 50%), and RAxML bootstrap support values (ML ≥ 50%) were shown at the nodes (BI/MP/ML). Tree length = 827, CI = 0.71, RI = 0.93, RC = 0.65, HI = 0.30. Ex-type isolates are in bold. Circles indicate isolates from fruits, and triangles indicate isolates from leaves.

**Figure 4 jof-08-00313-f004:**
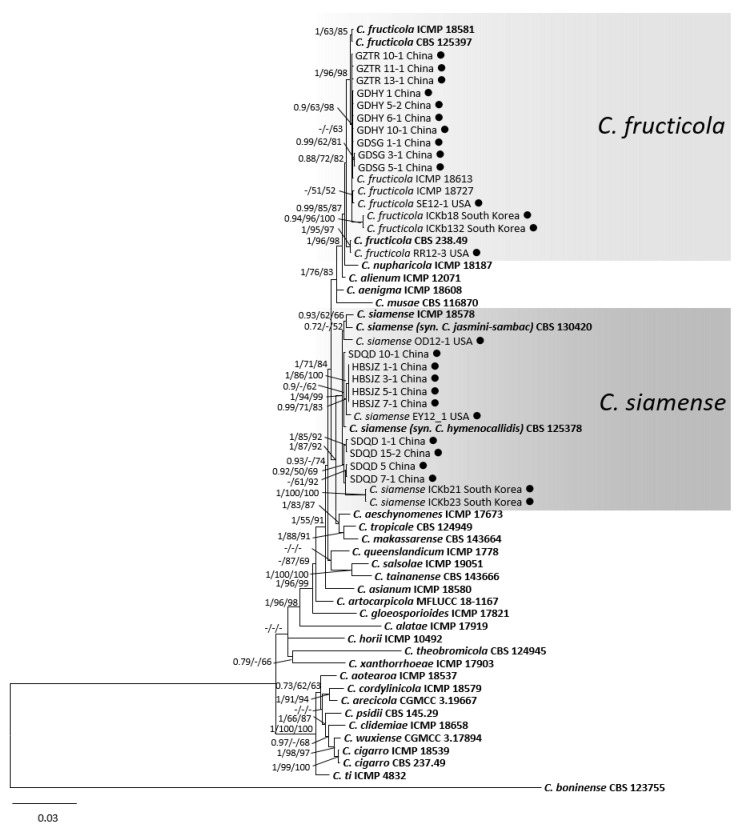
A Bayesian inference phylogenetic tree of 61 isolates in the *C. gloeosporioides* species complex. *C. boninense* (CBS 123755) was used as the outgroup. The tree was built using combined sequences of the *ACT*, *CAL*, *CHS-1*, *GAPDH*, ITS, and *TUB2*. BI posterior probability values (BI ≥ 0.70), MP bootstrap support values (MP ≥ 50%), and RAxML bootstrap support values (ML ≥ 50%) were shown at the nodes (BI/MP/ML). Tree length = 1303, CI = 0.76, RI = 0.84, RC = 0.63, HI = 0.24. Ex-type strains are in bold. Circles indicate isolates from fruits, and triangles indicate isolates from leaves.

**Figure 5 jof-08-00313-f005:**
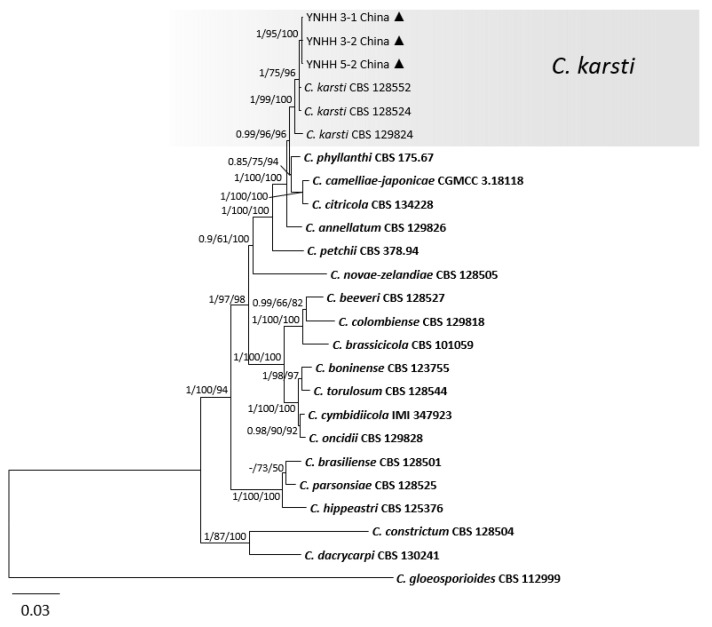
A Bayesian inference phylogenetic tree of 24 isolates in the *C. boninense* species complex. *C. gloeosporioides* (CBS 112999) was used as the outgroup. The tree was built using combined sequences of the ITS, *GAPDH*, *CHS-1*, *HIS3*, *ACT*, *TUB2* and *CAL*. BI posterior probability values (BI ≥ 0.70), MP bootstrap support values (MP ≥ 50%), and RAxML bootstrap support values (ML ≥ 50%) were shown at the nodes (BI/MP/ML). Tree length = 1404, CI = 0.76, RI = 0.79, RC = 0.60, HI = 0.24. Ex-type strains are in bold. Circles indicate isolates from fruits, and triangles indicate isolates from leaves.

**Figure 6 jof-08-00313-f006:**
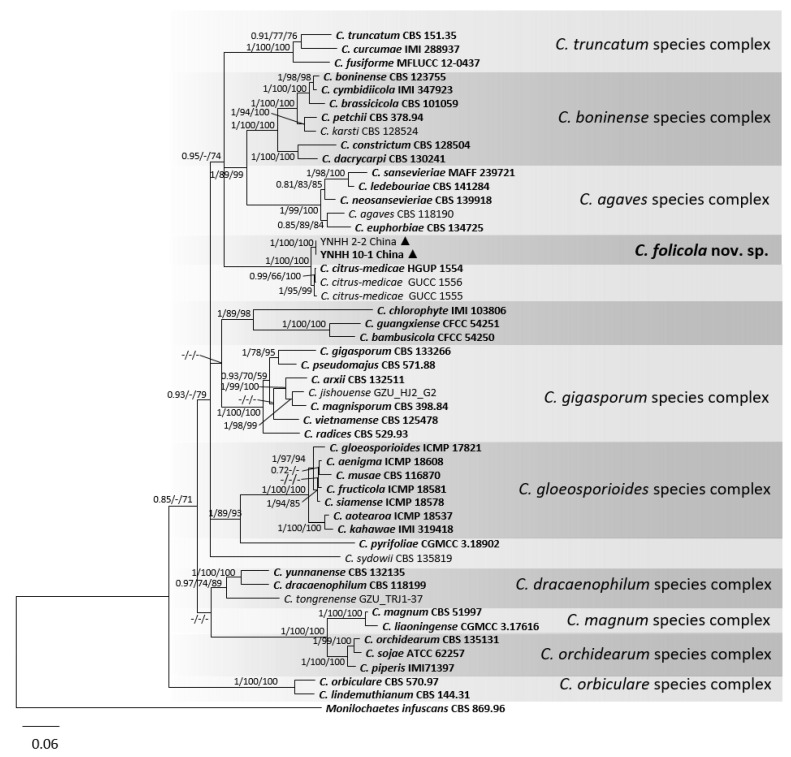
A Bayesian inference phylogenetic tree of 49 isolates of *Colletotrichum* spp. and outgroup. *Monilochaetes infuscans* (CBS 869.96) was used as the outgroup. The tree was built using combined sequences of the ITS, *GAPDH*, *CHS-1*, *ACT*, and *TUB2*. BI posterior probability values (BI ≥ 0.70), MP bootstrap support values (MP ≥ 50%), and RAxML bootstrap support values (ML ≥ 50%) were shown at the nodes (BI/MP/ML). Tree length = 4405, CI = 0.44, RI = 0.68, RC = 0.30, HI = 0.56. Ex-type strains are in bold. Circles indicate isolates from fruits, and triangles indicate isolates from leaves.

**Figure 7 jof-08-00313-f007:**
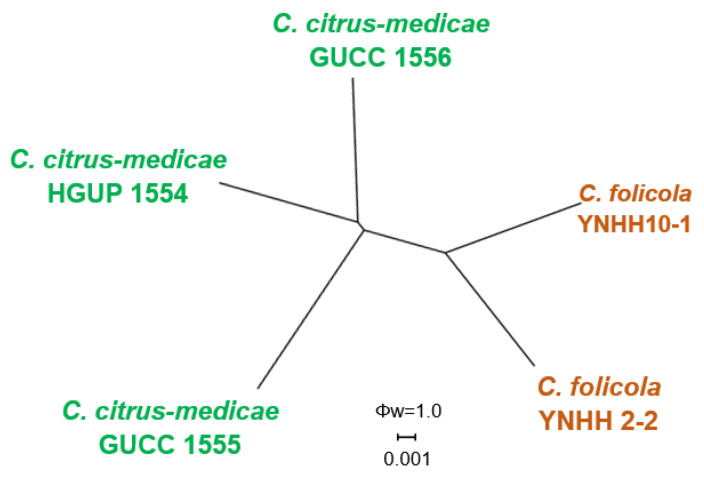
PHI test of *C. folicola* and phylogenetically related species using both LogDet transformation and splits decomposition. PHI test value (Φw) < 0.05 indicate significant recombination within the datasets.

**Figure 8 jof-08-00313-f008:**
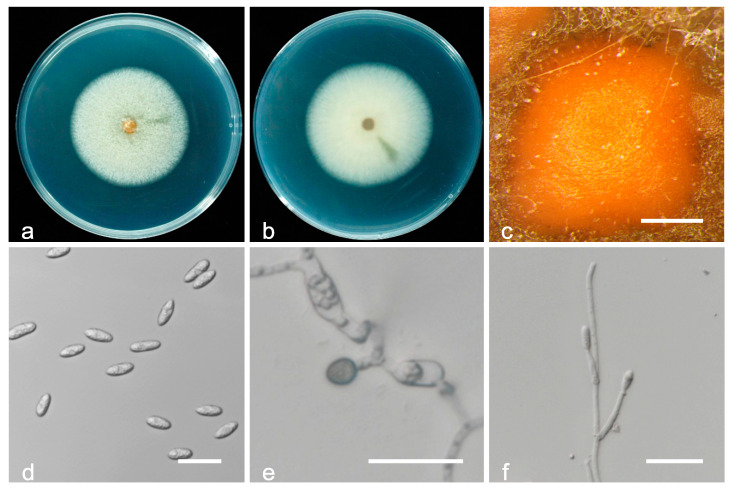
Biological characteristics of *Colletotrichum nymphaeae*. (**a**,**b**) Front and back view of six-day-old PDA culture; (**c**) conidiomata; (**d**) conidia; (**e**) appressoria; (**f**) conidiophores ((**a**–**e**) isolate HBYC 1; (**f**) isolate SCCD 1). Scale bars: (**c**) = 200 μm; (**d**–**f**) = 20 μm.

**Figure 9 jof-08-00313-f009:**
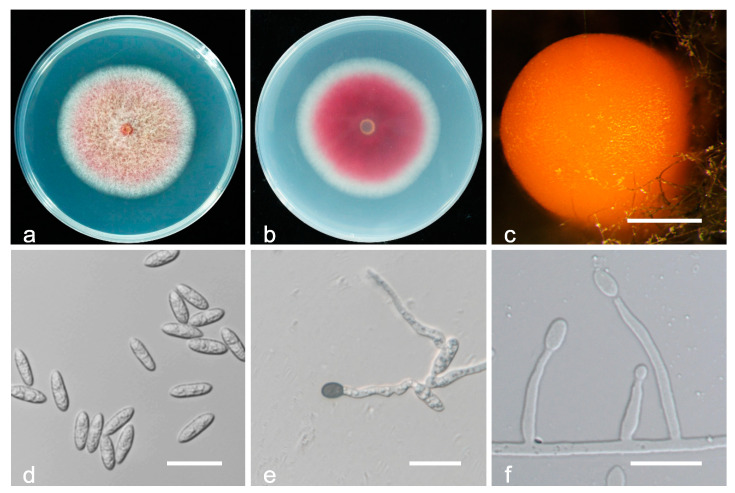
Biological characteristics of *Colletotrichum fioriniae*. (**a**,**b**) Front and back view of six-day-old PDA culture; (**c**) conidiomata; (**d**) conidia; (**e**) appressoria; (**f**) conidiophores ((**a**–**e**) isolate JXJA 6; (**f**) isolate JXJA 1). Scale bars: (**c**) = 200 μm; (**d**–**f**) = 20 μm.

**Figure 10 jof-08-00313-f010:**
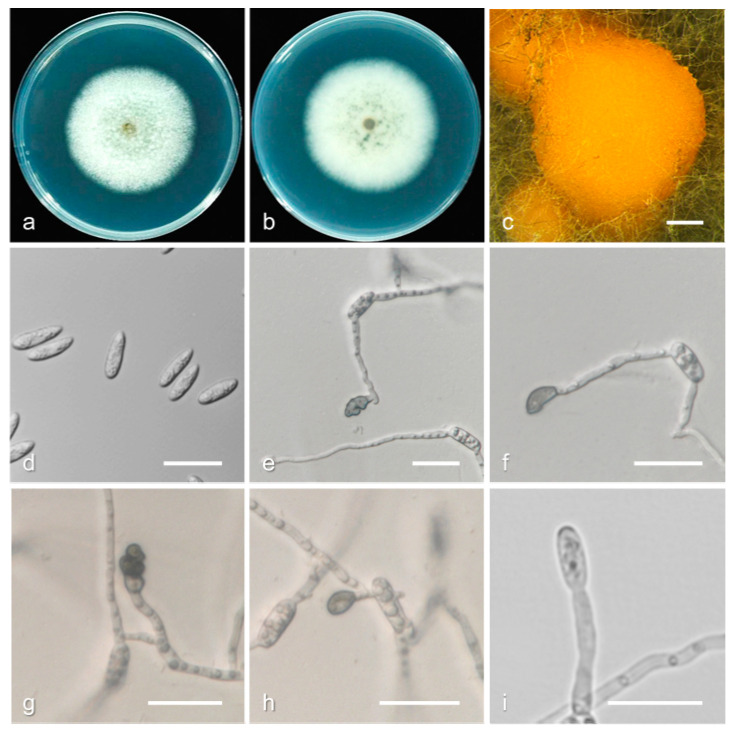
Biological characteristics of *Colletotrichum godetiae*. (**a**,**b**) Front and back view of six-day-old PDA culture; (**c**) conidiomata; (**d**) conidia; (**e**–**h**) appressoria; (**i**) conidiophores ((**a**–**f**,**i**) isolate YNHH 1-1, (**g**,**h**) YNHH 9-1). Scale bars: (**c**) = 200 μm; (**d**–**i**) = 20 μm.

**Figure 11 jof-08-00313-f011:**
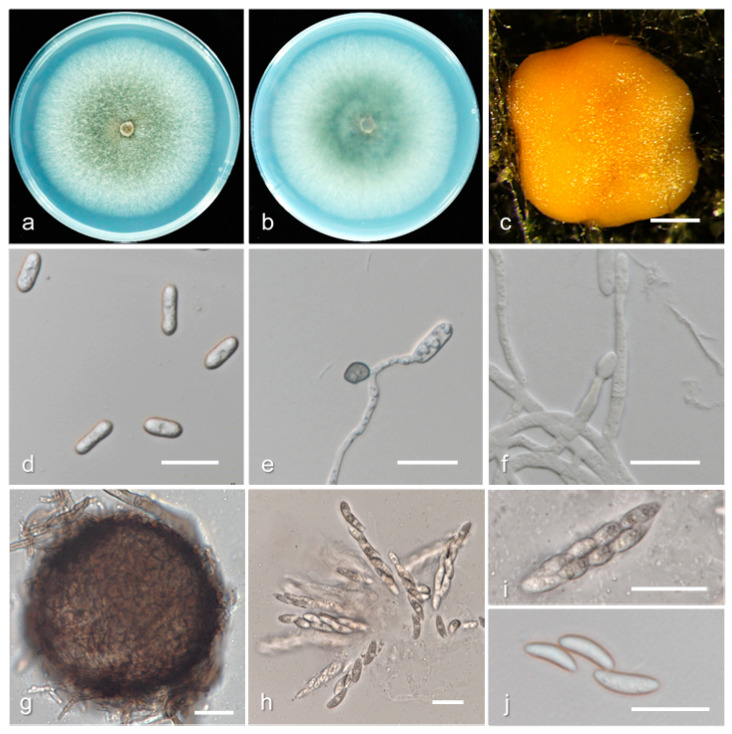
Biological characteristics of *Colletotrichum fructicola*. (**a**,**b**) Front and back view of six-day-old PDA culture; (**c**) conidiomata; (**d**) conidia; (**e**) appressoria; (**f**) conidiophores; (**g**) ascomata; (**h**,**i**) asci; (**j**) ascospores ((**a**–**e**) isolate GDHY 10-1; (**f**–**j**) isolate GDSG 1-1). Scale bars: (**c**) = 200 μm; (**d**–**j**) = 20 μm.

**Figure 12 jof-08-00313-f012:**
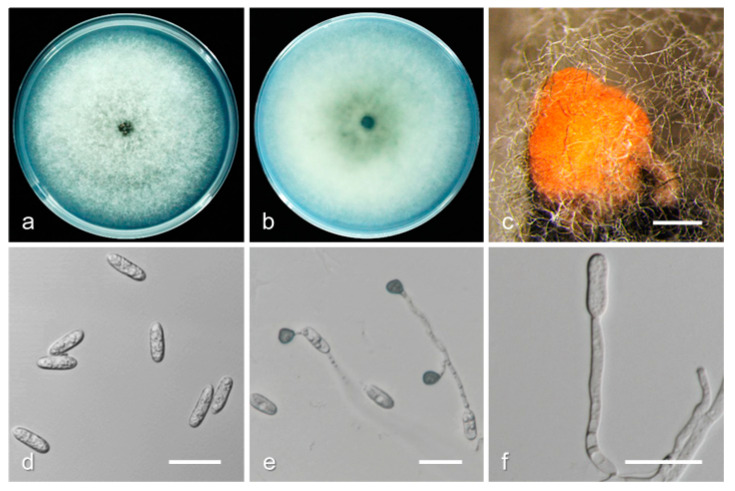
Biological characteristics of *Colletotrichum siamense*. (**a**,**b**) Front and back view of six-day-old PDA culture; (**c**) conidiomata; (**d**) conidia; (**e**) appressoria; (**f**) conidiophores ((**a**–**e**) isolate SDQD10-1; (**f**) isolate HBSJZ 1-1). Scale bars: (**c**) = 200 μm; (**d**–**f**) = 20 μm.

**Figure 13 jof-08-00313-f013:**
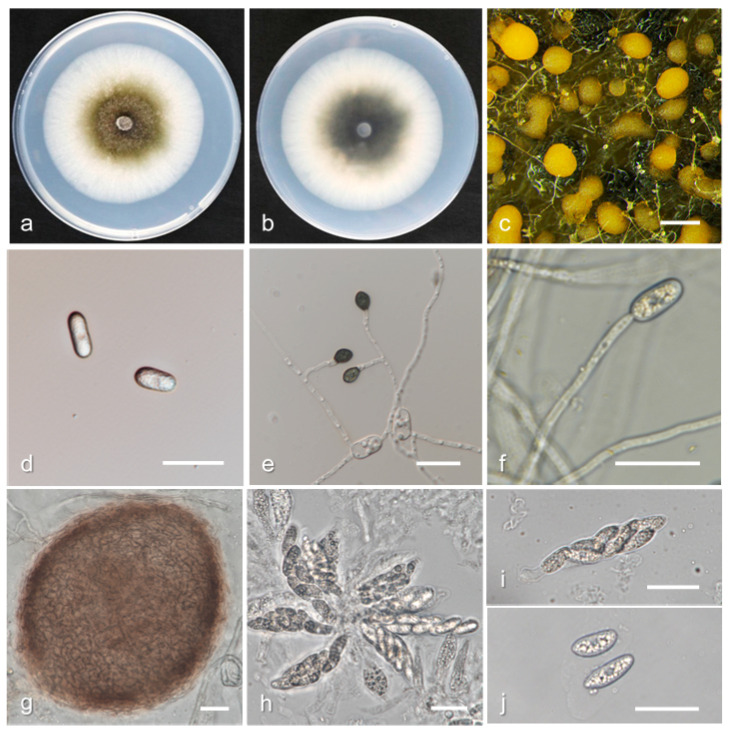
Biological characteristics of *Colletotrichum karsti*. (**a**,**b**) Front and back view of six-day-old PDA culture; (**c**) conidiomata; (**d**) conidia; (**e**) appressoria; (**f**) conidiophores; (**g**) ascomata; (**h**,**i**) asci; (**j**) ascospores ((**a**–**j**) isolate YNHH 3-1). Scale bars: (**c**) = 200 μm; (**d**–**j**) = 20 μm.

**Figure 14 jof-08-00313-f014:**
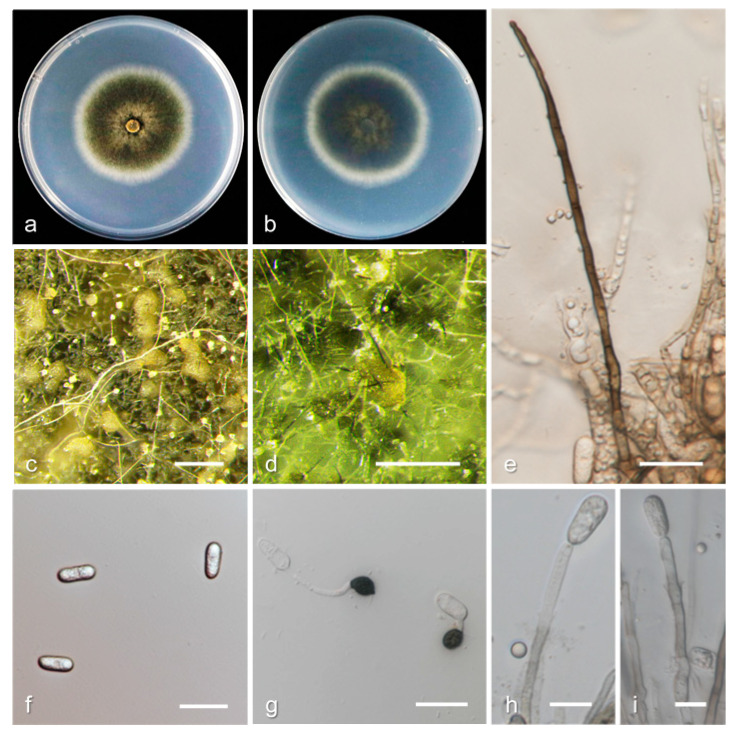
Biological characteristics of *Colletotrichum folicola*. (**a**,**b**) Front and back view of six-day-old PDA culture; (**c**,**d**) conidiomata; (**e**) setae; (**f**) conidia; (**g**) appressoria; (**h**) conidiophores ((**a**–**h**) isolate YNHH 10-1). Scale bars: (**c**,**d**) = 200 μm; (**e**–**g**) = 20 μm; (**h**,**i**) = 10 μm.

**Figure 15 jof-08-00313-f015:**
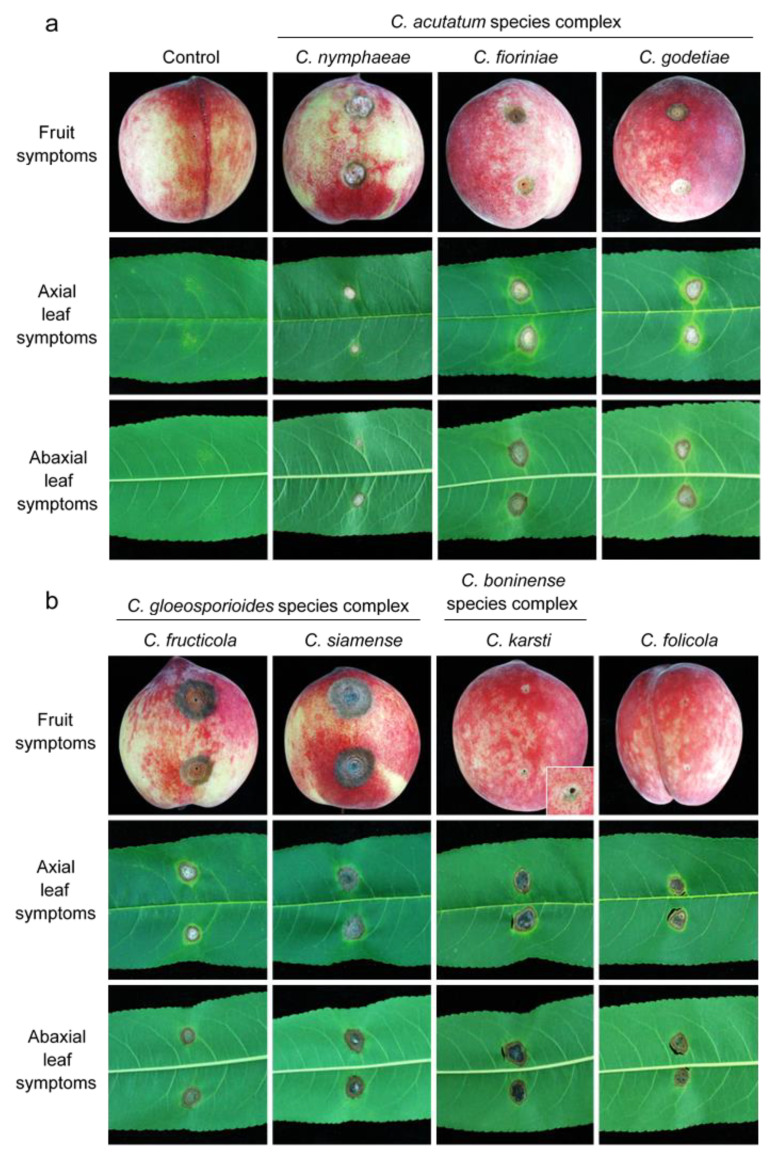
Symptoms of peach fruits and leaves induced by inoculation of spore suspensions of seven *Colletotrichum* spp. after six days at 25 °C. (**a**) Symptoms resulting from H_2_O, isolates HBYC 1, JXJA 6, and YNHH 1-1 (**left** to **right**). (**b**) Symptoms resulting from isolates GDHY 10-1, SDQD 10-1, YNHH3-1, and YNHH10-1 (**left** to **right**).

**Figure 16 jof-08-00313-f016:**
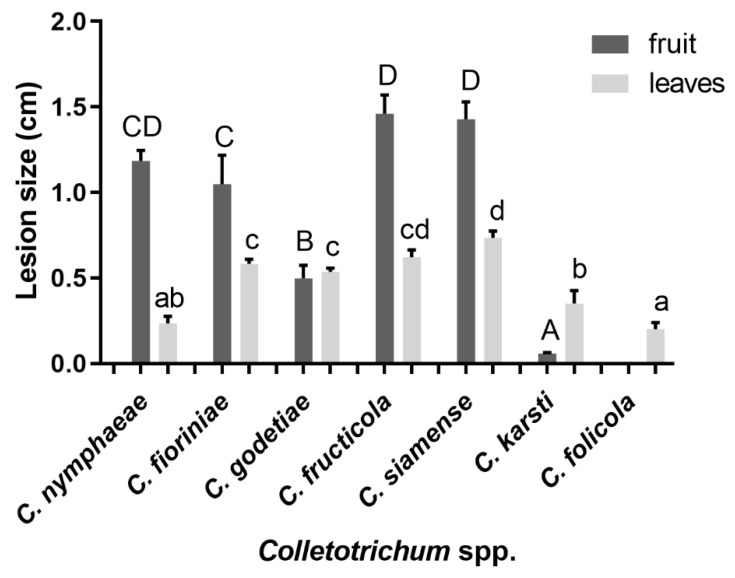
Lesion size on peach fruit and leaves of seven *Colletotrichum* spp. in the six days after inoculation. *C. nymphaeae* isolates FJFZ 1, HBJM 1-1, HBWH 3-2, HBYC 1, SCCD 1; *C. fioriniae* isolates GZTR 7-1, JXJA 1, JXJA 6, ZJLS 1, ZJLS 11-1; *C. godetiae* isolates YNHH 1-1, YNHH 2-1, YNHH 4-1, YNHH 7-2, YNHH 9-1; *C. fructicola* isolates GDHY 10-1, GDSG 1-1, GDSG 5-1, GZTR 10-1, GZTR 13-1; *C. siamense* isolates HBSJZ 1-1, HBSJZ 3-1, HBSJZ 5-1, HBSJZ 7-1, SDQD 10-1; *C. karsti* isolates YNHH 3-1, YNHH 3-2, YNHH 5-2; *C. folicola* isolates YNHH 2-2, YNHH 10-1. Letters over the error bars indicate a significant difference at the *p* = 0.05 level. Capital letters refer to fruit and lowercase letters to leaves.

**Table 1 jof-08-00313-t001:** Strains used for the phylogenetic analysis of *Colletotrichum* spp. and other species with details about host, location, and GenBank accession numbers.

Species	Culture ^a^	Host	Location	GenBank Accession Number						
				ITS	*GAPDH*	*CHS-1*	*ACT*	*HIS3*	*TUB2*	*CAL*
*C. acerbum*	CBS 128530 *	*Malus domestica*	New Zealand	JQ948459	JQ948790	JQ949120	JQ949780	JQ949450	JQ950110	-
*C. acutatum*	CBS 112996 *	*Carica papaya*	Australia	JQ005776	JQ948677	JQ005797	JQ005839	JQ005818	JQ005860	-
	C-1	*Prunus persica*	China	KX611163	KY049983	-	KY049982	-	KY049984	-
*C. aenigma*	ICMP 18608 *	*Persea americana*	Israel	JX010244	JX010044	JX009774	JX009443	-	JX010389	JX009683
*C. aeschynomenes*	ICMP 17673 *	*Aeschynomene virginica*	USA	JX010176	JX009930	JX009799	JX009483	-	JX010392	JX009721
*C. agaves*	CBS 118190	*Agave striate*	Mexico	DQ286221	-	-	-	-	-	-
*C. alatae*	ICMP 17919 *	*Dioscorea alata*	India	JX010190	JX009990	JX009837	JX009471	-	JX010383	JX009738
*C. alienum*	ICMP 12071 *	*Malus domestica*	New Zealand	JX010251	JX010028	JX009882	JX009572	-	JX010411	JX009654
*C. annellatum*	CBS 129826 *	*Hevea brasiliensis*	Colombia	JQ005222	JQ005309	JQ005396	JQ005570	JQ005483	JQ005656	JQ005743
*C. aotearoa*	ICMP 18537 *	*Coprosma* sp.	New Zealand	JX010205	JX010005	JX009853	JX009564	-	JX010420	JX009611
*C. arecicola*	CGMCC 3.19667 *	*Areca catechu*	China	MK914635	MK935455	MK935541	MK935374	-	MK935498	-
*C. artocarpicola*	MFLUCC 18-1167 *	*Artocarpus heterophyllus*	Thailand	MN415991	MN435568	MN435569	MN435570	-	MN435567	-
*C. arxii*	CBS 132511 *	*Paphiopedilum* sp.	Germany	KF687716	KF687843	KF687780	KF687802	-	KF687881	-
*C. asianum*	ICMP 18580 *	*Coffea arabica*	Thailand	FJ972612	JX010053	JX009867	JX009584	-	JX010406	FJ917506
*C. australe*	CBS 116478 *	*Trachycarpus fortunei*	South Africa	JQ948455	JQ948786	JQ949116	JQ949776	JQ949446	JQ950106	-
*C.* *bambusicola*	CFCC 54250 *	*Phyllostachys edulis*	China	MT199632	MT192844	MT192871	MT188638	-	MT192817	-
*C. beeveri*	CBS 128527 *	*Brachyglottis repanda*	New Zealand	JQ005171	JQ005258	JQ005345	JQ005519	JQ005432	JQ005605	JQ005692
*C. boninense*	CBS 123755 *	*Crinum asiaticum* var. *sinicum*	Japan	JQ005153	JQ005240	JQ005327	JQ005501	JQ005414	JQ005588	JQ005674
*C. brasiliense*	CBS 128501 *	*Passiflora edulis*	Brazil	JQ005235	JQ005322	JQ005409	JQ005583	JQ005496	JQ005669	JQ005756
*C. brassicicola*	CBS 101059 *	*Brassica oleracea* var. *gemmifera*	New Zealand	JQ005172	JQ005259	JQ005346	JQ005520	JQ005433	JQ005606	JQ005693
*C. brisbanense*	CBS 292.67 *	*Capsicum annuum*	Australia	JQ948291	JQ948621	JQ948952	JQ949612	JQ949282	JQ949942	-
*C. cairnsense*	CBS 140847 *	*Capsicum annuum*	Australia	KU923672	KU923704	KU923710	KU923716	KU923722	KU923688	-
*C. camelliae-japonicae*	CGMCC 3.18118 *	*Camellia japonica*	Japan	KX853165	KX893584	-	KX893576	-	KX893580	-
*C. chlorophyti*	IMI 103806 *	*Chlorophytum* sp.	India	GU227894	GU228286	GU228384	GU227992	-	GU228188	-
*C. chrysanthemi*	IMI 364540	*Chrysanthemum coronarium*	China	JQ948273	JQ948603	JQ948934	JQ949594	JQ949264	JQ949924	-
*C. ciggaro*	ICMP 18539 *	*Olea europaea*	Australia	JX010230	JX009966	JX009800	JX009523	-	JX010434	JX009635
	CBS 237.49 *	*Hypericum perforatum*	Germany	JX010238	JX010042	JX009840	JX009450	-	JX010432	JX009636
*C. citricola*	CBS 134228 *	*Citrus unshiu*	China	KC293576	KC293736	-	KC293616	-	KC293656	KC293696
*C. citrus-medicae*	HGUP 1554 *, GUCC 1554	*Citrus medica*	China	MN959910	MT006331	MT006328	MT006325	MT006334	-	-
	GUCC 1555	*Citrus medica*	China	MN959911	MT006332	MT006329	MT006326	MT006335	-	-
	GUCC 1556	*Citrus medica*	China	MN959912	MT006333	MT006330	MT006327	MT006336	-	-
*C. clidemiae*	ICMP 18658 *	*Clidemia hirta*	USA	JX010265	JX009989	JX009877	JX009537	-	JX010438	JX009645
*C. colombiense*	CBS 129818 *	*Passiflora edulis*	Colombia	JQ005174	JQ005261	JQ005348	JQ005522	JQ005435	JQ005608	JQ005695
*C. constrictum*	CBS 128504 *	*Citrus limon*	New Zealand	JQ005238	JQ005325	JQ005412	JQ005586	JQ005499	JQ005672	JQ005759
*C. cordylinicola*	ICMP 18579 *	*Cordyline fruticosa*	Thailand	JX010226	JX009975	JX009864	HM470235	-	JX010440	HM470238
*C. curcumae*	IMI 288937 *	*Curcuma longa*	India	GU227893	GU228285	GU228383	GU227991	-	GU228187	-
*C. cuscutae*	IMI 304802 *	*Cuscuta* sp.	Dominica	JQ948195	JQ948525	JQ948856	JQ949516	JQ949186	JQ949846	-
*C. cymbidiicola*	IMI 347923 *	*Cymbidium* sp.	Australia	JQ005166	JQ005253	JQ005340	JQ005514	JQ005427	JQ005600	JQ005687
*C. dacrycarpi*	CBS 130241 *	*Dacrycarpus dacrydioides*	New Zealand	JQ005236	JQ005323	JQ005410	JQ005584	JQ005497	JQ005670	JQ005757
*C. dracaenophilum*	CBS 118199 *	*Dracaena* sp.	China	JX519222	JX546707	JX519230	JX519238	-	JX519247	-
*C. eriobotryae*	BCRC FU31138 *	*Eriobotrya japonica*	China	MF772487	MF795423	MN191653	MN191648	MN19168	MF795428	-
*C. euphorbiae*	CBS 134725 *	*Euphorbia* sp.	South Africa	KF777146	KF777131	KF777128	KF777125		KF777247	-
*C. fioriniae*	CBS 128517 *	*Fiorinia externa*	USA	JQ948292	JQ948622	JQ948953	JQ949613	JQ949283	JQ949943	-
	IMI 324996	*Malus pumila*	USA	JQ948301	JQ948631	JQ948962	JQ949622	JQ949292	JQ949952	-
	CBS 126526	*Primula* sp.	Netherlands	JQ948323	JQ948653	JQ948984	JQ949644	JQ949314	JQ949974	-
	CBS 124958	*Pyrus* sp.	USA	JQ948306	JQ948636	JQ948967	JQ949627	JQ949297	JQ949957	-
	CBS 119292	*Vaccinium* sp.	New Zealand	JQ948313	JQ948643	JQ948974	JQ949634	JQ949304	JQ949964	-
	ICKb31	*Prunus persica*	South Korea	LC516639	LC516653	LC516660	-	-	LC516646	-
	ICKb36	*Prunus persica*	South Korea	LC516640	LC516654	LC516661	-	-	LC516647	-
	ICKb47	*Prunus persica*	South Korea	LC516641	LC516655	LC516662	-	-	LC516648	-
	C.2.4.2	*Prunus persica*	USA	KX066091	KX066094	-	-	-	KX066088	-
	CaEY12_1	*Prunus persica*	USA	KX066093	KX066096	-	-	-	KX066090	-
*C. fructicola*	ICMP 18581 *	*Coffea arabica*	Thailand	JX010165	JX010033	JX009866	FJ907426	-	JX010405	-
	ICMP 18613	*Limonium sinuatum*	Israel	JX010167	JX009998	JX009772	JX009491	-	JX010388	JX009675
	ICMP 18581 *	*Coffea arabica*	Thailand	JX010165	JX010033	JX009866	FJ907426	-	JX010405	FJ917508
	ICMP 18727	*Fragaria* × *ananassa*	USA	JX010179	JX010035	JX009812	JX009565	-	JX010394	JX009682
	CBS 125397 *	*Tetragastris panamensis*	Panama	JX010173	JX010032	JX009874	JX009581	-	JX010409	JX009674
	CBS 238.49 *	*Ficus edulis*	Germany	JX010181	JX009923	JX009839	JX009495	-	JX010400	JX009671
	ICKb18	*Prunus persica*	South Korea	LC516635	LC516649	LC516656	-	-	LC516642	LC516663
	ICKb132	*Prunus persica*	South Korea	LC516636	LC516650	LC516657	-	-	LC516643	LC516664
	RR12-3	*Prunus persica*	USA	-	KJ769247	-	-	-	KM245092	KJ769239
	SE12-1	*Prunus persica*	USA	-	KJ769248	-	-	-	-	KJ769237
*C. fusiforme*	MFLUCC 12– 0437 *	*unknown*	Thailand	KT290266	KT290255	KT290253	KT290251	-	KT290256	-
*C. gigasporum*	CBS 133266 *	*Centella asiatica*	Madagascar	KF687715	KF687822	KF687761	-	-	KF687866	-
*C. gloeosporioides*	CBS 112999 *	*Citrus sinensis*	Italy	JQ005152	JQ005239	JQ005326	JQ005500	JQ005413	JQ005587	JQ005673
	ICMP 17821 *	*Citrus sinensis*	Italy	JX010152	JX010056	JX009818	JX009531	-	JX010445	JX009731
*C. godetiae*	CBS 796.72	*Aeschynomene virginica*	USA	JQ948407	JQ948738	JQ949068	JQ949728	JQ949398	JQ950058	-
	CBS 133.44 *	*Clarkia hybrida*	Denmark	JQ948402	JQ948733	JQ949063	JQ949723	JQ949393	JQ950053	-
	IMI 351248	*Ceanothus* sp.	UK	JQ948433	JQ948764	JQ949094	JQ949754	JQ949424	JQ950084	-
*C. guangxiense*	CFCC 54251 *	*Phyllostachys edulis*	China	MT199633	MT192834	MT192861	MT188628	-	MT192805	-
*C. hippeastri*	CBS 125376 *	*Hippeastrum vittatum*	China	JQ005231	JQ005318	JQ005405	JQ005579	JQ005492	JQ005665	JQ005752
*C. horii*	ICMP 10492 *	*Diospyros kaki*	Japan	GQ329690	GQ329681	JX009752	JX009438		JX010450	JX009604
*C. indonesiense*	CBS 127551 *	*Eucalyptus* sp.	Indonesia	JQ948288	JQ948618	JQ948949	JQ949609	JQ949279	JQ949939	-
*C. javanense*	CBS 144963 *	*Capsicum annuum*	Indonesia	MH846576	MH846572	MH846573	MH846575	-	MH846574	-
*C. jishouense*	GZU_HJ2_G2	*Nothapodytes pittosporoides*	China	MH482931	MH681657	-	MH708134	-	MH727472	-
*C. johnstonii*	CBS 128532 *	*Solanum lycopersicum*	New Zealand	JQ948444	JQ948775	JQ949105	JQ949765	JQ949435	JQ950095	-
*C. kahawae*	IMI 319418 *	*Coffea arabica*	Kenya	JX010231	JX010012	JX009813	JX009452	-	JX010444	-
*C. karsti*	CBS 128524	*Citrullus lanatus*	New Zealand	JQ005195	JQ005282	JQ005369	JQ005543	JQ005456	JQ005629	JQ005716
	CBS 129824	*Musa* AAA	Colombia	JQ005215	JQ005302	JQ005389	JQ005563	JQ005476	JQ005649	JQ005736
	CBS 128552	*Synsepalum dulcificum*	Taiwan	JQ005188	JQ005275	JQ005362	JQ005536	JQ005449	JQ005622	JQ005709
*C. laticiphilum*	CBS 112989 *	*Hevea brasiliensis*	India	JQ948289	JQ948619	JQ948950	JQ949610	JQ949280	JQ949940	-
*C. ledebouriae*	CBS 141284 *	*Ledebouria floridunda*	South Africa	KX228254	-	-	KX228357	-	-	-
*C. liaoningense*	CGMCC 3.17616 *	*Capsicum sp.*	China	KP890104	KP890135	KP890127	KP890097	-	KP890111	-
*C. limetticola*	CBS 114.14 *	*Citrus aurantifolia*	USA	JQ948193	JQ948523	JQ948854	JQ949514	JQ949184	JQ949844	-
*C. lindemuthianum*	CBS 144.31 *	*Phaseolus vulgaris*	Germany	JQ005779	JX546712	JQ005800	JQ005842	-	JQ005863	-
*C. magnisporum*	CBS 398.84 *	unknown	unknown	KF687718	KF687842	KF687782	KF687803	-	KF687882	-
*C. magnum*	CBS 519.97 *	*Citrullus lanatus*	USA	MG600769	MG600829	MG600875	MG600973	-	MG601036	-
*C. makassarense*	CBS 143664 *	*Capsicum annuum*	Indonesia	MH728812	MH728820	MH805850	MH781480	-	MH846563	-
*C. musae*	CBS 116870 *	*Musa* sp.	USA	JX010146	JX010050	JX009896	JX009433	-	HQ596280	JX009742
*C. neosansevieriae*	CBS 139918 *	*Sansevieria trifasciata*	South Africa	KR476747	KR476791	-	KR476790	-	KR476797	-
*C. novae-zelandiae*	CBS 128505 *	*Capsicum annuum*	New Zealand	JQ005228	JQ005315	JQ005402	JQ005576	JQ005489	JQ005662	JQ005749
*C. nupharicola*	ICMP 18187 *	*Nuphar lutea* subsp.*polysepala*	USA	JX010187	JX009972	JX009835	JX009437	-	JX010398	JX009663
*C. nymphaeae*	CBS 515.78 *	*Nymphaea alba*	Netherlands	JQ948197	JQ948527	JQ948858	JQ949518	JQ949188	JQ949848	-
	CBS 130.80	*Anemone* sp.	Italy	JQ948226	JQ948556	JQ948887	JQ949547	JQ949217	JQ949877	-
	IMI 360386	*Pelargonium graveolens*	India	JQ948206	JQ948536	JQ948867	JQ949527	JQ949197	JQ949857	-
	CBS 125973	*Fragaria × ananassa*	UK	JQ948232	JQ948562	JQ948893	JQ949553	JQ949223	JQ949883	-
	CaC04_42	*Prunus persica*	USA	KX066092	KX066095	-	-	-	KX066089	-
	PrpCnSC13–01	*Prunus persica*	Brazil	MK761066	MK770424	MK770421	-	-	MK770427	-
	PrpCnSC13–02	*Prunus persica*	Brazil	MK765508	MK770425	MK770422	-	-	MK770428	-
	PrpCnSC13–10	*Prunus persica*	Brazil	MK765507	MK770426	MK770423	-	-	MK770429	-
*C. oncidii*	CBS 129828 *	*Oncidium* sp.	Germany	JQ005169	JQ005256	JQ005343	JQ005517	JQ005430	JQ005603	JQ005690
*C. orbiculare*	CBS 570.97 *	*Cucumis sativus*	Europe	KF178466	KF178490	KF178515	KF178563	-	KF178587	-
*C. orchidearum*	CBS 135131 *	*Dendrobium nobile*	Netherlands	MG600738	MG600800	MG600855	MG600944	-	MG601005	-
*C. orchidophilum*	CBS 632.80 *	*Dendrobium* sp.	USA	JQ948151	JQ948481	JQ948812	JQ949472	JQ949142	JQ949802	-
*C. parsonsiae*	CBS 128525 *	*Parsonsia capsularis*	New Zealand	JQ005233	JQ005320	JQ005407	JQ005581	JQ005494	JQ005667	JQ005754
*C. paxtonii*	IMI 165753 *	*Musa* sp.	Saint Lucia	JQ948285	JQ948615	JQ948946	JQ949606	JQ949276	JQ949936	-
*C. petchii*	CBS 378.94 *	*Dracaena marginata*	Italy	JQ005223	JQ005310	JQ005397	JQ005571	JQ005484	JQ005657	JQ005744
*C. phormii*	CBS 118194 *	*Phormium* sp.	Germany	JQ948446	JQ948777	JQ949107	JQ949767	JQ949437	JQ950097	-
*C. phyllanthi*	CBS 175.67 *	*Phyllanthus acidus*	India	JQ005221	JQ005308	JQ005395	JQ005569	JQ005482	JQ005655	JQ005742
*C. piperis*	IMI 71397 *	*Piper nigrum*	Malaysia	MG600760	MG600820	MG600867	MG600964	-	MG601027	-
*C. pseudomajus*	CBS 571.88 *	*Camellia sinensis*	China	KF687722	KF687826	KF687779	KF687801	-	KF687883	-
*C. psidii*	CBS 145.29 *	*Psidium* sp.	Italy	JX010219	JX009967	JX009901	JX009515	-	JX010443	JX009743
*C. pyricola*	CBS 128531 *	*Pyrus communis*	New Zealand	JQ948445	JQ948776	JQ949106	JQ949766	JQ949436	JQ950096	-
*C. pyrifoliae*	CGMCC 3.18902 *	*Pyrus pyrifolia*	China	MG748078	MG747996	MG747914	MG747768	-	MG748158	-
*C. queenslandicum*	ICMP 1778 *	*Carica papaya*	Australia	JX010276	JX009934	JX009899	JX009447	-	JX010414	JX009691
*C. radicis*	CBS 529.93 *	unknown	Costa Rica	KF687719	KF687825	KF687762	KF687785	-	KF687869	-
*C. salicis*	CBS 607.94 *	*Salix* sp.	Netherlands	JQ948460	JQ948791	JQ949121	JQ949781	JQ949451	JQ950111	-
*C. salsolae*	ICMP 19051 *	*Salsola tragus*	Hungary	JX010242	JX009916	JX009863	JX009562	-	JX010403	JX009696
*C. sansevieriae*	MAFF 239721 *	*Sansevieria trifasciata*	Japan	AB212991	-	-	-	-	-	-
*C. scovillei*	CBS 1265299 *	*Capsicum* sp.	Indonesia	JQ948267	JQ948597	JQ948928	JQ949588	JQ949258	JQ949918	-
*C. siamense*	ICMP 18578 *, MFLU 090230	*Coffea arabica*	Thailand	JX010171	JX009924	JX009865	FJ907423	-	JX010404	FJ917505
*C. siamense (syn. C. hymenocallidis)*	CBS 125378 *	*Hymenocallis americana*	China	JX010278	JX010019	GQ856730	GQ856775	-	JX010410	JX009709
*C. siamense (syn. C. jasmini-sambac)*	CBS 130420 *	*Jasminum sambac*	Vietnam	HM131511	HM131497	JX009895	HM131507	-	JX010415	JX009713
	ICKb21	*Prunus persica*	South Korea	LC516637	LC516651	LC516658	-	-	LC516644	LC516665
	ICKb23	*Prunus persica*	South Korea	LC516638	LC516652	LC516659	-	-	LC516645	LC516666
	OD12-1	*Prunus persica*	USA	-	KJ769240	-	-	-	KM245089	KJ769234
	EY12-1	*Prunus persica*	USA	-	KJ769246	-	-	-	KM245086	KJ769236
*C. simmondsii*	CBS 122122 *	*Carica papaya*	Australia	JQ948276	JQ948606	JQ948937	JQ949597	JQ949267	JQ949927	-
*C. sloanei*	IMI 364297 *	*Theobroma cacao*	Malaysia	JQ948287	JQ948617	JQ948948	JQ949608	JQ949278	JQ949938	-
*C. sojae*	ATCC 62257 *	*Glycine max*	USA	MG600749	MG600810	MG600860	MG600954	-	MG601016	-
*C. sydowii*	CBS 135819	*Sambucus* sp.	China	KY263783	KY263785	KY263787	KY263791	-	KY263793	-
*C. tainanense*	CBS 143666 *	*Capsicum annuum*	Taiwan	MH728818	MH728823	MH805845	MH781475	-	MH846558	-
*C. theobromicola*	CBS 124945 *	*Theobroma cacao*	Panama	JX010294	JX010006	JX009869	JX009444	-	JX010447	JX009591
*C. ti*	ICMP 4832 *	*Cordyline* sp.	New Zealand	JX010269	JX009952	JX009898	JX009520	-	JX010442	JX009649
*C. tongrenense*	GZU_TRJ1-37	*Nothapodytes pittosporoides*	China	MH482933	MH705332	-	MH717074	-	MH729805	-
*C. torulosum*	CBS 128544 *	*Solanum melongena*	New Zealand	JQ005164	JQ005251	JQ005338	JQ005512	JQ005425	JQ005598	JQ005685
*C. trichellum*	CBS 217.64 *	*Hedera helix*	UK	GU227812	GU228204	GU228302	GU227910	-	GU228106	-
*C. tropicale*	CBS 124949 *	*Theobroma cacao*	Panama	JX010264	JX010007	JX009870	JX009489	-	JX010407	JX009719
*C. truncatum*	CBS 151.35 *	*Phaseolus lunatus*	USA	GU227862	GU228254	GU228352	GU227960	-	GU228156	-
*C. vietnamense*	CBS 125478 *	*Coffea* sp.	Vietnam	KF687721	KF687832	KF687769	KF687792	-	KF687877	-
*C. walleri*	CBS 125472 *	*Coffea* sp.	Vietnam	JQ948275	JQ948605	JQ948936	JQ949596	JQ949266	JQ949926	-
*C. wanningense*	CGMCC 3.18936 *	*Hevea brasiliensis*	China	MG830462	MG830318	MG830302	MG830270	-	MG830286	-
*C. wuxiense*	CGMCC 3.17894 *	*Camellia sinensis*	China	KU251591	KU252045	KU251939	KU251672	-	KU252200	KU251833
*C. xanthorrhoeae*	ICMP 17903 *	*Xanthorrhoea preissii*	Australia	JX010261	JX009927	JX009823	JX009478	-	JX010448	JX009653
*C. yunnanense*	CBS 132135 *	*Buxus* sp.	China	JX546804	JX546706	JX519231	JX519239	-	JX519248	-
*Monilochaetes infuscans*	CBS 869.96 *	*Ipomoea batatas*	South Africa	JQ005780	JX546612	JQ005801	JQ005843	-	JQ005864	-

^a^ CBS: Culture collection of the Centraalbureau voor Schimmelcultures; ICMP: International Collection of Microorganisms from Plants, Auckland, New Zealand; CGMCC: China General Microbiological Culture Collection; MFLUCC: Mae Fah Luang University Culture Collection, Chiang Rai, Thailand; IMI: Culture collection of CABI Europe UK Centre, Egham, UK; BCRC: Bioresource Collection and Research Center, Hsinchu, Taiwan; MFLU: Herbarium of Mae Fah Luang University, Chiang Rai, Thailand; MAFF: MAFF Genebank Project, Ministry of Agriculture, Forestry and Fisheries, Tsukuba, Japan; ATCC: American Type Culture Collection. * = Ex-holotype or ex-epitype cultures.

**Table 2 jof-08-00313-t002:** Comparison of alignment properties in parsimony analyses of gene/locus and nucleotide substitution models used in phylogenetic analyses of *C. acutatum* species complex.

Gene/Locus	ITS	*GAPDH*	*CHS-1*	*HIS3*	*ACT*	*TUB2*	Combined
No. of taxa	72	72	68	60	63	72	72
Aligned length (with gaps)	546	265	282	387	248	492	2240
Invariable characters	501	152	244	289	170	374	1750
Uninformative variable characters	26	56	13	32	30	60	217
Phylogenetically informative characters	19	57	25	66	48	58	273
Tree length (TL)	59	176	64	190	117	165	827
Consistency index (CI)	0.85	0.80	0.73	0.66	0.75	0.79	0.71
Retention index (RI)	0.97	0.95	0.94	0.93	0.94	0.94	0.93
Rescaled consistency index (RC)	0.82	0.76	0.69	0.61	0.71	0.75	0.65
Homoplasy index (HI)	0.15	0.20	0.27	0.34	0.25	0.21	0.30
Nucleotide substitution model	HKY + I	HKY + G	K80 + I	GTR + I + G	GTR + G	GTR + G	GTR + I + G

**Table 3 jof-08-00313-t003:** Comparison of alignment properties in parsimony analyses of gene/locus and nucleotide substitution models used in phylogenetic analyses of *C. gloeosporioides* species complex.

Gene/Locus	*ACT*	*CAL*	*CHS-1*	*GAPDH*	ITS	*TUB2*	Combined
No. of taxa	54	58	58	62	58	61	62
Aligned length (with gaps)	314	744	300	307	614	735	3034
Invariable characters	232	520	239	154	555	489	2209
Uninformative variable characters	54	139	22	77	36	156	484
Phylogenetically informative characters	28	85	39	76	23	90	341
Tree length (TL)	115	324	102	264	78	349	1303
Consistency index (CI)	0.84	0.83	0.69	0.75	0.81	0.83	0.76
Retention index (RI)	0.85	0.92	0.84	0.84	0.87	0.87	0.84
Rescaled consistency index (RC)	0.71	0.76	0.58	0.63	0.70	0.72	0.63
Homoplasy index (HI)	0.17	0.17	0.31	0.25	0.19	0.17	0.24
Nucleotide substitution model	HKY + G	GTR + G	K80 + G	HKY + I	SYM + I + G	HKY + I	GTR + I + G

**Table 4 jof-08-00313-t004:** Comparison of alignment properties in parsimony analyses of gene/locus and nucleotide substitution models used in phylogenetic analyses of *C. boninense* species complex.

Gene/Locus	ITS	*GAPDH*	*CHS-1*	*HIS3*	*ACT*	*TUB2*	*CAL*	Combined
No. of taxa	25	25	23	23	25	25	24	25
Aligned length (with gaps)	553	286	280	393	276	502	449	2763
Invariable characters	489	120	224	295	174	348	259	1932
Uninformative variable characters	40	82	25	28	53	75	103	408
Phylogenetically informative characters	24	84	31	70	49	79	87	423
Tree length (TL)	87	286	89	210	164	237	300	1404
Consistency index (CI)	0.86	0.80	0.76	0.66	0.82	0.75	0.80	0.76
Retention index (RI)	0.88	0.79	0.79	0.79	0.83	0.75	0.85	0.79
Rescaled consistency index (RC)	0.75	0.64	0.60	0.52	0.68	0.56	0.70	0.60
Homoplasy index (HI)	0.14	0.20	0.24	0.34	0.18	0.25	0.18	0.24
Nucleotide substitution model	SYM + I + G	HKY + I	K80 + G	GTR + I + G	GTR + G	HKY + I	HKY + G	GTR + I + G

**Table 5 jof-08-00313-t005:** Comparison of alignment properties in parsimony analyses of gene/locus and nucleotide substitution models used in phylogenetic analyses of *C. folicola* and other taxa.

Gene/Locus	ITS	*GAPDH*	*CHS-1*	*ACT*	*TUB2*	combined
No. of taxa	50	47	44	47	44	50
Aligned length (with gaps)	571	321	265	279	529	1981
Invariable characters	367	63	163	102	223	934
Uninformative variable characters	53	21	20	39	50	183
Phylogenetically informative characters	151	237	82	138	256	864
Tree length (TL)	630	1312	389	671	1300	4405
Consistency index (CI)	0.51	0.44	0.41	0.48	0.44	0.44
Retention index (RI)	0.76	0.68	0.66	0.71	0.67	0.68
Rescaled consistency index (RC)	0.39	0.30	0.27	0.34	0.30	0.30
Homoplasy index (HI)	0.49	0.56	0.59	0.53	0.56	0.56
Nucleotide substitution model	GTR + I + G	HKY + I + G	GTR + I + G	HKY + I + G	HKY + I + G	GTR + I + G

**Table 6 jof-08-00313-t006:** A list of all *Colletotrichum* isolates collected from peaches in China based on preliminary identification.

Species	Location	Host	Number of Isolates	Date	Daily Mean Temperature (°C) ^a^
*C. fioriniae*	Lishui, Zhejiang	Juicy peach, Yanhong, fruit	17	14 September 2017	29
	Tongren, Guizhou	Juicy peach, fruit	14	8 August 2018	29
	Jian, Jiangxi	Yellow peach, fruit	6	21 August 2018	31
*C. folicola*	Honghe, Yunnan	Winter peach, Hongxue, leaf	2	17 August 2017	26
*C. fructicola*	Heyuan, Guangdong	Juicy peach, fruit	19	28 June 2017	29
	Shaoguan, Guangdong	Juicy peach, Yingzui, fruit	10	3 August 2018	30
	Tongren, Guizhou	Juicy peach, fruit	10	8 August 2018	29
*C. godetiae*	Honghe, Yunnan	Winter peach, Hongxue, leaf	15	17 August 2017	26
*C. karstii*	Honghe, Yunnan	Winter peach, Hongxue, leaf	3	17 August 2017	26
*C. nymphaeae*	Yichang, Hubei	Yellow peach, NJC83, fruit	11	30 April 2017	19
	Jingmen, Hubei	Yellow peach, NJC83, fruit	14	25 April 2017	18
	Jingmen, Hubei	Juicy peach, Chunmi, fruit	11	25 April 2017	18
	Wuhan, Hubei	Juicy peach, Zaoxianhong, fruit	17	18 April 2017	20
	Wuhan, Hubei	Flat peach, Zaoyoupan, fruit	12	18 April 2017	20
	Wuhan, Hubei	Juicy peach, leaft	9	14 June 2017	25
	Xiaogan, Hubei	Juicy peach, Chunmei, fruit	4	10 May 2017	20
	Qingzhen, Guizhou	Juicy peach, Yingqing, fruit	8	21 August 2017	24
	Tongren, Guizhou	Juicy peach, fruit	2	08 August 2018	29
	Guilin, Guangxi	Juicy peach, Chunmi, fruit	38	18 May 2018	25
	Guilin, Guangxi	Juicy peach, Chunmi, leaf	4	18 May 2018	25
	Fuzhou, Fujian	Yellow peach, huangjinmi, fruit	12	27 July 2018	31
	Chengdu, Sichuan	Yellow peach, Zhongtaojinmi, fruit	7	28 June 2018	26
*C. siamense*	Qingdao, Shandong	Juicy peach, Yangjiaomi, fruit	27	22 August 2017	27
	Shijiazhuang, Hebei	Juicy peach, Dajiubao, fruit	14	3 August 2018	30
Total			286		

^a^ The average of the daily mean temperatures on the sampling day and the previous six days.

## Data Availability

Alignments generated during the current study are available from TreeBASE (http://treebase.org/treebase-web/home.html; study 29227). All sequence data are available in the NCBI GenBank, following the accession numbers in the manuscript.

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
