# Peer review of "Colletotrichum* Species Associated with Peaches in China"

_jof, 2022, doi:10.3390/jof8030313_

Round 1
Reviewer 1 Report
The manuscript “Colletotrichum Species Associated with Peach in China” is a very interesting paper that up-date the knowledge on the population dynamics of the peach anthracnose causing pathogens, bearing in mind the current taxonomic framework in Colletotrichum as well as the host range, organ specificity, and geographic distribution of each species, in order to foresee future population dynamics and to anticipate better informed protection strategies.
Beyond some minor comments include in the manuscript file I suggest that the authors could include in the abstract the hypothesis raised concerning the fungicide tolerance of C. acutatum species complex and its distribution in peach production areas in China, since this demonstrates the importance of a sustainable agriculture based on informed decisions and its impact in the pathogen population dynamics.
Considering the experimental design, it will be interesting to know the infection rates of the Colletotrichum spp. isolates inoculated on peach fruit without wounding tissues once this information could be important to guess the impact of the disease in the field.

Author Response
To Reviewer 1:
The manuscript “Colletotrichum Species Associated with Peach in China” is a very interesting paper that up-date the knowledge on the population dynamics of the peach anthracnose causing pathogens, bearing in mind the current taxonomic framework in Colletotrichum as well as the host range, organ specificity, and geographic distribution of each species, in order to foresee future population dynamics and to anticipate better informed protection strategies.
Response: Thank you very much for your positive comments.
Beyond some minor comments include in the manuscript file I suggest that the authors could include in the abstract the hypothesis raised concerning the fungicide tolerance of C. acutatum species complex and its distribution in peach production areas in China, since this demonstrates the importance of a sustainable agriculture based on informed decisions and its impact in the pathogen population dynamics.
Response: Thank you for your kind suggestion. We inserted " C. nymphaeae is the most prevalent species of Colletotrichum on peach in China, which may be the result of fungicide selection." on line 22 of the first page. We did not describe the distribution and fungicide resistance of the entire C. acutatum species complex, and only describe C. nymphaeae belonging to C. acutatum species complex. C. nymphaeae is the most prevalent species in China.
Considering the experimental design, it will be interesting to know the infection rates of the Colletotrichum spp. isolates inoculated on peach fruit without wounding tissues once this information could be important to guess the impact of the disease in the field.
Response: In nature Colletotrichum spp. also infect host through wounds and stomata (De Silva et al. 2017), so the experimental design matches this fact. In addition, different peach and leaves may have small invisible wounds, which will affect the accuracy of the experimental results. We stabbed wounds to make the susceptibility of all tissues closer to get more accurate results.
Below are some editorial suggestions to improve the manuscript:
- Page 3
Line 121: Although the authors referred table 1 as preliminary results, it is a result so it should not be included in material and methods!
Response: We moved table 1 to page 18, line 268, which becomes table 6 according to your suggestion.
Line 137: As indicated above Table 1 should not be in this section. E.g. C. folicola is the newly described species (accordingly with the abstract), so why it is in the material and methods?
Response: As indicated above we moved table 1 to the results according to your suggestion.
- Page 4
Line 149: Please take care with scientific number notation
Response: Thank you for your correction, we replaced “105” with “105”.
Line 160-162: The primers pairs and the proper reference could be included as supplementary material.
Response: We replaced “ITS1-F/ITS-4[27,28], GDF1/GDR1 [29], CHS-79F/CHS-345R [30], ACT-512F/ACT-783R [30], Btub2Fd/Btub4Rd [31] or T1/Bt2b [32,33], CYLH3F/CYLH3R [34], and CL1C/CL2C [35], respectively” by “described in Table S1”, and listed the sequences and references of primers in supplementary material table S1 according to your suggestion.
Line 163-165: The same concerning the PCR conditions! The PCR conditions could be included in a supplementary table with the primers pairs.
Response: We listed the annealing temperatures in supplementary material table S1 according to your suggestion. We modified these sentences as “The PCR conditions were 4 min at 95℃, followed by 35 cycles of 95℃ for 30 s, annealing for 30 s at different temperature for different genes/loci (Table S1), and 72℃ for 45 s, with a final extension at 72℃ for 7 min. DNA sequencing was performed at Tianyi Huiyuan Biotechnology Co., Ltd (Wuhan, China) with an ABI 3730XL sequencer from ThermoFisher” .
- Page 34
Line 742: ?? (Control?)
Response: We replaced “CK” with “Control” according to your suggestion.
Reviewer 2 Report
The manuscript provides interesting new information on the presence of distribution of the genus Colletotrichum in China, a very important pathogen of agricultural crops. The article is well written, the methods are adequate and the results clearly presented.
However, I think the paper could be improved by considering the following:
- The manuscript is very long. Several data sets (e.g. Tables 2, 8 and 9) don’t have to be included in the main article, but should be moved to ‘supplemental data’)
- The authors describe a new species, C. folicula. However, this species is not mentioned in the title or abstract, and just briefly in the discussion.
- The locations from which the microbial strains were isolated should be described in more details. Only for samples from Yunnan Province some environmental information was given. Were all the peaches collected from commercial orchards or also private garden etc.? Were all the peaches treated with fungicides or were some grown organically? The area from which isolates were collected (Figure 2) is huge, therefore some crucial climatic conditions should be mentioned (differences in temperature and/or rainfall?). Were the orchards irrigated or dry-farmed?
- The description of the new species, C. folicola, should include more information on growth at different temperatures (minimum, maximum, optimum temperature?).
Author Response
To Reviewer 2:
The manuscript provides interesting new information on the presence of distribution of the genus Colletotrichum in China, a very important pathogen of agricultural crops. The article is well written, the methods are adequate and the results clearly presented.
Response: Thank you very much for your positive comments.
However, I think the paper could be improved by considering the following: The manuscript is very long. Several data sets (e.g. Tables 2, 8 and 9) don’t have to be included in the main article, but should be moved to ‘supplemental data’)
Response: We moved Tables 2, 8 and 9 to supplemental data as Tables S2, S3 and S4 according to your suggestion.
The authors describe a new species, C. folicola. However, this species is not mentioned in the title or abstract, and just briefly in the discussion.
Response: We inserted “sp. nov.” after “and one newly identified species, C. folicola” in the abstract (line 20 of the first page) according to your suggestion.
The locations from which the microbial strains were isolated should be described in more details. Only for samples from Yunnan Province some environmental information was given. Were all the peaches collected from commercial orchards or also private garden etc.? Were all the peaches treated with fungicides or were some grown organically? The area from which isolates were collected (Figure 2) is huge, therefore some crucial climatic conditions should be mentioned (differences in temperature and/or rainfall?). Were the orchards irrigated or dry-farmed?
Response: Thank you very much for your kind suggestion. In lines 125-127 of the third page, we replaced “16 locations” with “14 commercial peach orchards and two nurseries (Wuhan, Hubei and Fuzhou, Fujian)”, and inserted “which were dry-farmed and sprayed with fungicides for anthracnose control” after “in 11 provinces of China”. And we added daily mean temperature in table 6 (table 1 before revision, page 18, line 268 of revised manuscript) and changed the collection time from “Year” to “Date”. As for rainfall, the higher the humidity, the more severe the disease occurrence, which is same for all Colletotrichum spp. on peach. We only investigate the occurrence of different Colletotrichum species, not focus on the severity of the anthracnose.
The description of the new species, C. folicola, should include more information on growth at different temperatures (minimum, maximum, optimum temperature?).
Response: Actually, in taxonomy, the description of culture characteristics generally only provides the growth rate at one temperature. The daily mean temperature of C. folicola we collected was 26℃, and the ripening of peaches was in the hot summer. So we evaluated the adaptability of the species at 25℃ and 30℃.
Reviewer 3 Report
The study is original and of practical and scientific interest. The experimental design and the methods are appropriate. The results are clearly presented. Although rather speculative the conclusions are consistent with the results. The literature cited has to be completed with additional references as suggested (see notes in the text). In my opinion Table 8, which is very large and not essential, could be reported as a supplemental table. For other minor editing observations see notes in the text (attached file)

Author Response
To Reviewer 3:
The study is original and of practical and scientific interest. The experimental design and the methods are appropriate. The results are clearly presented.
Response: Thank you very much for your positive comments.
Although rather speculative the conclusions are consistent with the results. The literature cited has to be completed with additional references as suggested (see notes in the text).
Response: We revised the literature cited according to your suggestions (details are listed below).
In my opinion Table 8, which is very large and not essential, could be reported as a supplemental table.
Response: We moved Table 8 to supplementary material as Table S3.
Below are some editorial suggestions to improve the manuscript:
- Page 1
Line 45: Colletotrichum species
Response: We replaced “The fungi” with “Colletotrichum species” according to your suggestion.
- Page 2
Line 47: they
Response: We replaced “the fungi” with “they” according to your suggestion.
Line 99: infected
Response: We replaced “disease” with “infected” according to your suggestion.
- Page 3
Line 103: , many Colletotrichum species are polyphagous and multiple species may infect the same host plant [10,11.......]
[10] Cacciola, S.O.; Gilardi, G.; Faedda, R.; Schena, L.; Pane, A.; Garibaldi, A.; Gullino, M.L. Characterization of Colletotrichum ocimi Population Associated with Black Spot of Sweet Basil (Ocimum basilicum) in Northern Italy. Plants 2020, 9, 654. https://doi.org/10.3390/plants9050654
[11] Riolo, M.; Aloi, F.; Pane, A.; Cara, M.; Cacciola, S.O. Twig and Shoot Dieback of Citrus, a New Disease Caused by Colletotrichum Species. Cells 2021, 10, 449. https://doi.org/10.3390/cells10020449
Response: We inserted “many Colletotrichum species are polyphagous and multiple species may infect the same host plant” after “on environment conditions” according to your suggestion. And We added two citations [10] and [11], replaced “[10,11]” with “[10-13]”.
Line 108: [17-20]
[16] Adaskaveg, J. E., Hartin, R. J. Characterization of Colletotrichum acutatum isolates causing anthracnose of almond and peach in California. Phytopathology 1997 87, 979-987. https://doi.org/10.1094/PHYTO.1997.87.9.979
Response: We added one citation and replaced “[15-17]” with “[17-20]” according to your suggestion.
- Page 16
Line 228: used in
Response: We replaced “subjected for” with “used in” according to your suggestion.
Line 245: and the mean of two
The lesion size was determined as the mean of two perpendicular diameters
Response: We replaced “The lesion diameters were measured perpendicularly” with “The lesion size was determined as the mean of two perpendicular diameters” according to your suggestion.
- Page 17
Line 252: WHAT ABOUT THE SYMPTOMATIC TWIGS YOU SHOWED IN FIG. 1j?
Response: We tried isolating the Colletotrichum spp. from symptomatic twigs, but unfortunately we didn't get.
- Page 25
Line 490: recovered from
Response: We replaced “occurred on” with “recovered from” according to your suggestion.
Line 491: and Olea europaea [57]
[57] Schena, L.; Abdelfattah, A.; Mosca, S.; Li Destri Nicosia, M.G.; Agosteo, G.E.; Cacciola, S.O. Quantitative detection of Colletotrichum godetiae and C. acutatum sensu stricto in the phyllosphere and carposphere of olive during four phenological phases. Eur. J. Plant Pathol. 2017, 149, 337–347.
Response: We added one citation and inserted “and Olea europaea [49]” after “Citrus aurantium” according to your suggestion.
- Page 26
Line 543: wide
Response: Thank you for your correction, we replaced “wild” with “wide”.
- Page 27
Line 641: spp.
Response: We replaced “sp.” with “spp.” according to your suggestion.
Line 642: [11,...
Response: We added one citation and replaced “[39,56,57]” with “[11,33,52,53]” according to your suggestion.
- Page 30
Line 717: I SUGGEST TO INSERT AS A SUPPLEMETAL TABLE
Response: We moved Table 8 to supplementary material as Table S3 according to your suggestion.
- Page 35
Line 824: same
Response: We replaced “similar” with “same” according to your suggestion.
- Page 36
Line 874: seven
Response: We replaced “7” with “seven” according to your suggestion.
- Page 37
Line 914: [10] Cacciola, S.O.; Gilardi, G.; Faedda, R.; Schena, L.; Pane, A.; Garibaldi, A.; Gullino, M.L. Characterization of Colletotrichum ocimi Population Associated with Black Spot of Sweet Basil (Ocimum basilicum) in Northern Italy. Plants 2020, 9, 654. https://doi.org/10.3390/plants9050654
[11] Riolo, M.; Aloi, F.; Pane, A.; Cara, M.; Cacciola, S.O. Twig and Shoot Dieback of Citrus, a New Disease Caused by Colletotrichum Species. Cells 2021, 10, 449. https://doi.org/10.3390/cells10020449
Response: We did it accordingly.
Line 924: Adaskaveg, J. E., and Hartin, R. J. Characterization of Colletotrichum acutatum isolates causing anthracnose of almond and peach in California. Phytopathology 1997 87, 979-987. https://doi.org/10.1094/PHYTO.1997.87.9.979
Response: We inserted it.
- Page 39
Line 1000: [57] Schena, L.; Abdelfattah, A.; Mosca, S.; Li Destri Nicosia, M.G.; Agosteo, G.E.; Cacciola, S.O. Quantitative detection of Colletotrichum godetiae and C. acutatum sensu stricto in the phyllosphere and carposphere of olive during four phenological phases. Eur. J. Plant Pathol. 2017, 149, 337–347.
Response: We inserted it.